# Let's Split Up: Zero-Shot Classifier Edits for Fine-Grained Video Understanding

**Kaiting Liu, Hazel Doughty**
Leiden University

## Abstract

Video recognition models are typically trained on fixed taxonomies which are often too coarse, collapsing distinctions in object, manner or outcome under a single label. As tasks and definitions evolve, such models cannot accommodate emerging distinctions and collecting new annotations and retraining to accommodate such changes is costly. To address these challenges, we introduce *category splitting*, a new task where an existing classifier is edited to refine a coarse category into finer subcategories, while preserving accuracy elsewhere. We propose a zero-shot editing method that leverages the latent compositional structure of video classifiers to expose fine-grained distinctions without additional data. We further show that low-shot fine-tuning, while simple, is highly effective and benefits from our zero-shot initialization. Experiments on our new video benchmarks for category splitting demonstrate that our method substantially outperforms vision-language baselines, improving accuracy on the newly split categories without sacrificing performance on the rest. Project page: https://kaitingliu.github.io/Category-Splitting/

## 1 Introduction

Categorization underlies recognition in vision, yet most models assume a fixed taxonomy that rarely matches real-world complexity. A single label can cover many visually distinct cases, and as applications mature, new distinctions often become important. In video understanding, such refinements are especially common: subtle differences in motion, timing, or object interactions can completely change the meaning of an action. For example, the broad label *open* can mask distinctions by object (*open cupboard*), manner (*open by pushing*), speed (*open quickly*) or outcome (*open halfway*). These examples highlight that action categories are inherently multidimensional, yet current recognition models commit to a single, fixed partition.

A straightforward solution is to retrain the model with new annotations. Yet this is costly, requiring extensive labeled data and a full training cycle. Vision-language models (VLMs) (Radford et al., 2021; Zhao et al., 2024) appear to offer a shortcut, allowing new categories at test time via text prompts. However, VLMs rely on massive video-text corpora that are rarely available in specialized domains, and seldom capture the subtle temporal cues of fine-grained actions. Continual learning (Wang et al., 2024b) provides another angle, aiming to expand the label space without forgetting. However it assumes access to training data for each new class and targets entirely novel categories rather than considering relationships to existing ones. Current approaches therefore fall short when an existing category must be divided into finer subcategories with little or no supervision.

We address this challenge by introducing **category splitting** (Figure 1). The task is to edit an existing classifier to refine a coarse label into fine-grained subcategories, while preserving other predictions. We focus on actions, where fine-grained distinctions are common and challenging, often hinging on subtle motion or temporal cues. Our key observation is that modern video backbones capture latent structure that can be decomposed to separate fine-grained variations even without direct supervision. Building on this insight, we propose a zero-shot editing method to expose new subcategories without additional data. When limited supervision is available, simple low-shot fine-tuning proves highly effective, particularly when initialized from our zero-shot edit. Experiments on our new category splitting benchmarks SSv2-Split and FineGym-Split demonstrate substantial gains on newly split categories without sacrificing performance elsewhere, consistently outperforming VLM baselines.

In summary, we: (i) define the category splitting task, (ii) propose a zero-shot editing method, (iii) show that low-shot fine-tuning is effective and benefits from zero-shot initialization, (iv) introduce benchmarks and metrics for evaluation, and (v) analyze where category splitting succeeds and fails.

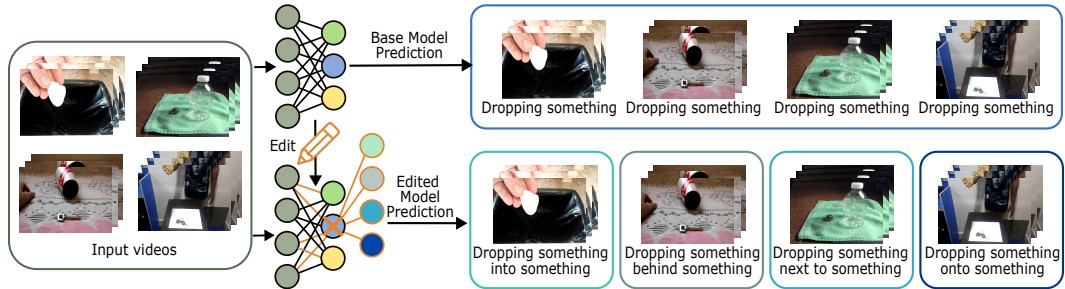

Figure 1: **Category splitting** aims to edit a trained video classifier by dividing a coarse label into multiple fine-grained subcategories, while keeping all other predictions unchanged. The challenge is to achieve this without retraining the full model and with zero or very few labels.

## 2 CATEGORY SPLITTING PROBLEM DEFINITION

We address the problem of *category splitting* as illustrated in Figure 1. The goal is to refine a chosen coarse category into finer-grained subcategories, while preserving the performance on others. Concretely, we start with a classification model trained to predict a set of categories, some of which may be more fine-grained while others are more coarse. We are given coarse-grained category within the current model that needs to be split into several new fine-grained subcategories. A naive approach would be to retrain the model from scratch with newly annotated data for these subcategories, together with the original training set. However, this is costly, requiring time-intensive data collection and annotation as well as a full training cycle. Our goal is to enable category splitting efficiently, operating with little or no labeled data. This allows rapid model adaptation in resource-constrained or time-sensitive scenarios and supports rare categories that would be absent from the training set.

Formally, let $\mathcal{X}$ be the input space, $\mathcal{Y}$ the label space, with the base classification model: $f_\theta : \mathcal{X} \rightarrow \mathcal{Y}$ Given coarse category $c \in \mathcal{Y}$, and its desired fine-grained subcategories $\mathcal{S}^c = \{s_1^c, s_2^c, \ldots, s_k^c\}$, the updated label space removes the coarse category and adds the fine-grained subcategories:

$$\mathcal{Y}' = (\mathcal{Y} \setminus \{c\}) \cup \mathcal{S}^c \quad \text{where } \mathcal{S}^c \cap \mathcal{Y} = \emptyset \tag{1}$$

We seek an editing method $E$, that locally and efficiently edits the original model $f_\theta$, yielding an updated model that supports fine-grained classification within the selected coarse category $c$:

$$E(f_\theta) = f_{\theta'}, \quad \text{where } f_{\theta'} : \mathcal{X} \rightarrow \mathcal{Y}' \tag{2}$$

We desire two properties for the updated model $f_{\theta'}$: generality and locality, adapted from model editing in NLP (Mitchell et al., 2022a). Generality means the model edits should extend beyond any given training samples and correctly classify unseen examples of the new subcategories $\mathcal{S}^c$. Locality means the edits should preserve predictions of all other existing model categories $\mathcal{Y} \setminus \{c\}$.

## 3 ZERO-SHOT CATEGORY SPLITTING

Fine-grained categories may lack annotated examples, as with rare events or anomalies. We therefore focus on zero-shot category splitting (Figure 2). Our key insight is that modern video backbones already encode rich latent features that capture compositional fine-grained variations within coarse classes, even without explicit compositional labels or text supervision. Exploiting this, we design a simple but effective approach that edits only the classification head, leaving the backbone unchanged. We view each fine-grained subcategory as a coarse category combined with a *modifier* that specifies a particular variation. This exposes a compositional structure within the classification head that we can exploit. Building on this, Section 3.1 introduces a retrieval-based method that first decomposes existing fine-grained categories into coarse concepts and modifiers, and then reuses suitable modifiers to split existing coarse categories into previously unseen fine-grained categories. Section 3.2 then extends this idea with a lightweight alignment model that maps textual modifier descriptions into the classification head weight space, enabling generalization to modifiers that do not appear in the original label set.

### 3.1 ZERO-SHOT EDITING: MODIFIER RETRIEVAL

We assume that the classifier we are editing is mixed-granularity video classifier, that is, it has been trained to predict a set of categories, some coarse and some fine. To split a selected coarse category,

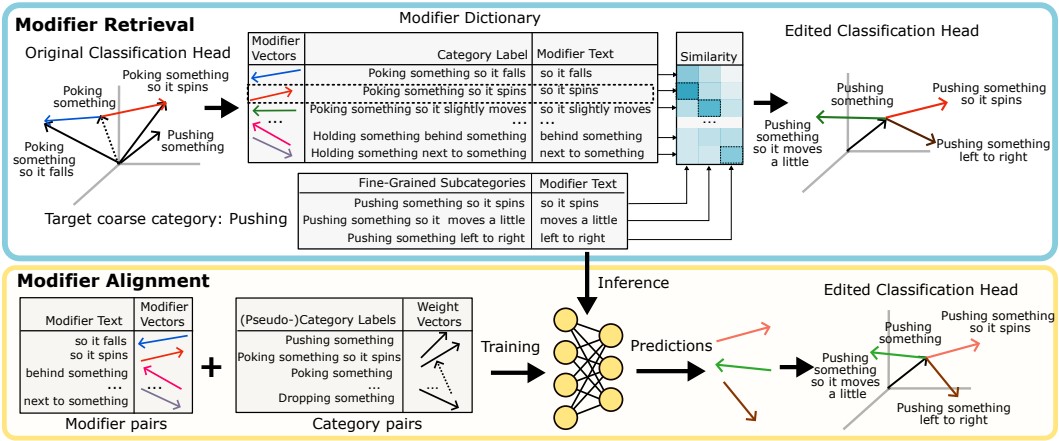

Figure 2: **Zero-shot Category Splitting**. Given a trained video classifier, our goal is to split a coarse category (e.g. *pushing*) into fine-grained subcategories without any video data. **Modifier retrieval** first exposes compositional structure in the video classifier's classification head from which it builds a dictionary of modifier vectors. The classifier is then edited by retrieving the appropriate modifier vector and adding it to the coarse category's weight vector to create a new fine-grained subcategory. To generalize to unseen modifiers, **modifier alignment** learns a lightweight mapping from modifier text to modifier vectors, using category text/weight vectors as additional supervision.

we build on the principle of compositionality: fine-grained concepts can be expressed as a coarse base concept combined with a modifier that specifies a particular variation. For example, *pushing left to right* can be viewed as the coarse action *pushing* composed with the directional modifier *left to right*. If the model already distinguishes between related classes such as *throwing left to right* and *throwing right to left*, the relevant modifier to split the *pushing* category is likely already encoded in the model. We exploit this by extracting modifier vectors from classifier weights, constructing a modifier dictionary and transferring the appropriate the modifiers to the coarse category being split.

**Modifier Dictionary Construction.** We extract modifiers by grouping existing categories in the classifier's label space $\mathcal{Y}$ that share a base concept, forming a pseudo coarse category $\tilde{c}$ with fine-grained variants $\mathcal{S}^{\tilde{c}} \subset \mathcal{Y}$. For example the set $\mathcal{S}^{\tilde{c}} = \{$*poking so it spins*, *poking so it falls*, *poking so it slightly moves*$\}$ would give a pseudo coarse category $\tilde{c}$ that corresponds to *poking*. Since the classifier already distinguishes between the fine-grained variants in $\mathcal{S}^{\tilde{c}}$, the differences between their weight vectors reflect the modifier directions encoded in the classification head $\theta_{head}$. To expose this structure, we construct explicit modifier vectors representing each fine-grained category's difference from the pseudo coarse cateory. Let $w_y$ denote the classifier weight vector of a category $y$. We approximate the weight vector of the pseudo coarse category $v_{\tilde{c}}$ as the mean of the associated fine-grained classifier weight vectors $w_y$:

$$v_{\tilde{c}} = \frac{1}{|\mathcal{S}^{\tilde{c}}|} \sum_{y \in \mathcal{S}^{\tilde{c}}} w_y. \tag{3}$$

The modifier vector $v_m$ for a fine-grained class $y \in \mathcal{S}^{\tilde{c}}$ is then obtained by subtracting this shared base component:

$$v_m = w_y - v_{\tilde{c}}. \tag{4}$$

We store these modifier vectors a dictionary $\mathcal{M}_{mod}$, each paired with a descriptive key. In this work, we use text labels as a convenient way to identify both base concepts and modifiers, through other forms of metadata such as attributes could also be used. Given the text description $t_y$ of a category $y \in \mathcal{Y}$, we identify the shared base description $t_{\tilde{c}}$ of its pseudo coarse category $\tilde{c}$. The modifier text is then $t_m = t_y - t_{\tilde{c}}$, describing the semantic difference (e.g. *left to right*). Each entry in the dictionary stores the triplet:

$$\mathcal{M}_{mod} = \{(t_m, t_y, v_m)\} \tag{5}$$

**Modifier Vector Transfer.** Our goal is to split a coarse category $c$ into new fine-grained subcategories $\mathcal{S}^c = \{s_1^c, s_2^c, ..., s_k^c\}$ without any video examples, by editing only the classifier head. Using our modifier dictionary $\mathcal{M}_{mod}$ we construct new classifier weight vectors for each new subcategory. We assume that the modifier vectors in $\mathcal{M}_{mod}$ are sufficiently disentangled from their original coarse concept to across categories. Under this assumption, for each subcategory $s_j^c$ we retrieve the

most appropriate modifier vector in $\mathcal{M}_{mod}$ that best distinguishes $s_j^c$ from the coarse category to be split $c$. To do so, we first derive the modifier text $t_m^*$ describing the difference between $s_j^c$ and $c$ and encode it with a text encoder $\phi$. Each modifier text $t_m$ in the dictionary $\mathcal{M}_{mod}$ is likewise embedded, and we retrieve most similarity entry using cosine similarity ($\mathrm{sim}(\cdot, \cdot)$) to give us the modifier vector $v_m^*$:

$$v_m^* = \underset{(t_m, v_m) \in \mathcal{M}_{mod}}{\arg\max} \quad \mathrm{sim}(\phi(t_m), \phi(t_m^*)), \tag{6}$$

However, the same modifier can have different visual effects depending on the coarse concept, e.g. *pushing left to right* is more visually similar to *pulling left to right* than to *looking left to right*. To account for this, we extend retrieval to consider both the modifier and the coarse concept together. Given the text descriptions of the full label $t_s^*$ and modifier $t_m^*$ for the target subcategory $s_j^c$, the corresponding modifier vector $v_m^*$ is retrieved as:

$$v_m^* = \underset{(t_y, t_m, v_m) \in \mathcal{M}_{mod}}{\arg\max} \quad \mathrm{sim}(\phi(t_y), \phi(t_s^*)) + \mathrm{sim}(\phi(t_m)), \phi(t_m^*)), \tag{7}$$

To add the new fine-grained subcategories $\mathcal{S}^c = \{s_1^c, ..., s_k^c\}$ to the model, we replace the original coarse class weight $w_c$ in the classification head $\theta_{head}$ with additional weights $\theta'_{head} = [w_{s_1^c}, \ldots, w_{s_k^c}]$. Each subcategory weight $w_{s_j^c}$ is formed by adding the retrieved modifier vector to the coarse category weight:

$$w_{s_j^c} = w_c + v_m^* \tag{8}$$

This local edit reuses modifier knowledge already present in the model, enabling recognition of new fine-grained categories while leaving the rest of the model unchanged. The entire process is zero-shot, requiring no annotated videos and no backbone retraining.

## 3.2 ZERO-SHOT EDITING: MODIFIER ALIGNMENT

While modifier retrieval enables zero-shot category splitting, it is limited to modifiers that already appear in the trained model's label space. To handle modifiers outside of our dictionary $\mathcal{M}_{mod}$, we introduce an alignment module that maps text embeddings directly into the classifier weight space. This mapping allows us to synthesize modifier vectors directly from text, enabling generalization to unseen modifiers. We define an alignment function $g_\psi : \mathbb{R}^n \rightarrow \mathbb{R}^m$ that projects text embeddings from the encoder $\phi$ to modifier vectors in the classifier weight space. Learning this mapping requires pairs linking modifier text $t_m$ to its corresponding vector $v_m$. Crucially, no video data is needed, keeping our method zero-shot, as our modifier dictionary $\mathcal{M}_{mod}$ naturally provides such supervision. From it, we obtain modifier-level pairs:

$$\mathcal{D}_{mod} = \{(\phi(t_m), v_m) \mid (t_m, t_y, v_m) \in \mathcal{M}_{mod}\}. \tag{9}$$

However, modifier pairs alone provide too little supervision to reliably learn a mapping from text to classifier vector space. To enrich the training signal, we also align the text embeddings of existing categories $y \in \mathcal{Y}$ and pseudo-coarse categories $\tilde{c} \in \mathcal{Y}_{pseudo}$ with their corresponding weight vectors:

$$\mathcal{D}_{cat} = \{(\phi(t_y), w_y) \mid y \in \mathcal{Y}\} \cup \{(\phi(t_{\tilde{c}}), v_{\tilde{c}}) \mid \tilde{c} \in \mathcal{Y}_{pseudo}\} \tag{10}$$

Here $t_y$ and $t_{\tilde{c}}$ are the textual descriptions of the category and pseudo-coarse category respectively and $w_y$ and $v_{\tilde{c}}$ are the corresponding weight vectors (Eq. 3). We train the alignment module $g_\psi$ using mean squared error over the combined supervision:

$$\mathcal{L}_{\mathrm{MSE}} = \sum_{(\phi(t), v) \in \mathcal{D}_{mod} \cup \mathcal{D}_{cat}} \left\| g_\psi(\phi(t)) - v \right\|_2^2. \tag{11}$$

During training, only the alignment parameters $\psi$ are updated; the classifier and text encoder remain frozen. At inference, given a modifier text $t_m^*$ for a target subcategory $s_j^c$, we generate its vector $v_m^* = g_\psi(\phi(t_m^*))$ and extend the classification head with $w_{s_j^c} = w_c + v_m^*$. This enables generalization to unseen modifiers while remaining entirely zero-shot and requiring no backbone updates.

## 4  LOW-SHOT CATEGORY SPLITTING

While our method enables zero-shot category splitting, in practice a small number of labeled examples may be available. We therefore study low-shot category splitting (Fig. 3), using only a few videos from new subcategories. We find fine-tuning surprisingly effective even with extremely limited data, and that performance improves further with initialization from our zero-shot method.

**Isolated Finetuning.** A challenge in category splitting is preserving performance on the original categories. To avoid disrupting unrelated classes, we fine-tune only the new subcategory weights, keeping the rest of the model frozen. Let the original video classifier be $f_\theta$ with parameters $\theta =$

$\{\theta_{backbone}, \theta_{head}\}$, corresponding to the parameters of the backbone and the classification head. Given a coarse category $c$ encoded in $\theta_{head}$, suppose we wish to split $c$ into fine-grained subcategories $\mathcal{S}^c = \{s_1^c, \ldots, s_k^c\}$. We construct a small fine-tuning dataset $\mathcal{D}_{ft} = \{(x_i, y_i)\}_{i=1}^N$, where $y_i \in \mathcal{S}^c$. We focus on the extreme low-shot case with $N=1$ example per subcategory. To accommodate the expanded label space, we remove the coarse weight $w_c$ from the classifier head $\theta_{head}$ and replace it with a new set of subcategory weights $\theta'_{head} = [w_{s_1^c}, \ldots, w_{s_k^c}]$. We initialize each new subcategory weight $w_{s_j^c}$ from $w_c$, since the subcategories are subtle variations of their parent. The resulting model is $f_{\theta'}$, with parameters $\theta' = \{\theta_{backbone}, \theta_{head}, \theta'_{head}\}$. Fine-tuning is then applied only to $\theta'_{head}$ using cross entropy loss.

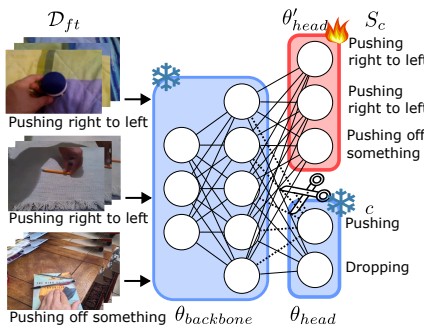

Figure 3: **Low-shot Category Splitting**. We edit the model by replacing the coarse category $c$ with $\theta'_{head}$. This head is fine-tuned with as little as one video per fine-grained subcategory, initialized with our zero-shot approach.

**Zero-Shot Initialization.** While isolated fine-tuning is a strong baseline for low-shot category splitting, it can be further improved with our zero-shot approach. For each fine-grained category $s_j^c$, we initialize the classifier weight as $w_{s_j^c} = w_c + v_m^*$, where $v_m^*$ can either be obtained from our modifier retrieval (Sec. 3.1) or modifier alignment (Sec. 3.2). This hybrid strategy combines model's latent fine-grained structure with available labeled samples, improving performance in the low-shot regime.

## 5 CATEGORY SPLITTING DATASETS

Since no benchmarks exist for category splitting, we construct two from Something-Something V2 (SSV2) (Goyal et al., 2017) and FineGym288 (Shao et al., 2020). SSV2 is a large-scale action dataset with 220K videos spanning 174 fine-grained categories. These categories capture subtle differences in object interactions and motion and can be naturally grouped into coarser semantic concepts. FineGym288 is a fine-grained gymnastics dataset with 31K videos annotated with 288 action categories organized hierarchically, providing a natural structure for evaluating category splitting.

**Constructing Coarse and Fine Labels.** We group the original fine-grained categories into semantically coherent coarse categories. For SSV2 we do this manually, for FineGym288 we follow the dataset's action hierarchy. The resulting coarse categories can be divided into fine-grained variants that differ in subtle aspects such as spatial relation, motion type, intensity, completion, antonyms, repetitions or body pose. A complete list of groupings is provided in Appendix A.

**Mixed Granularity Base Model.** Our task requires base models trained under mixed supervision, with some categories coarse and others fine. This mirrors real-world annotation settings where label granularity is inconsistent and creates the conditions to evaluate category splitting. Coarse categories serve as targets to be split, while fine-grained categories expose the backbone to subtle distinctions elsewhere in label space. To construct this setup, we partition coarse categories into two sets, A and B (see Appendix A) and define two complementary subsets. In subset A, categories in set A are collapsed into coarse labels, while those in set B retain their fine-grained labels. In subset B, the roles are reversed. This design ensures that every category is evaluated as a coarse label to be split. Each subset is used to train a base model, following the original SSV2/FineGym288 train and validation partition, which is then used to evaluate category splitting.

**Dataset Statistics.** Table 1 summarizes our benchmarks SSv2-Split and FineGym-Split. SSv2-Split contains 54 coarse categories and FineGym-Split 42, each split into 2-19 subcategories. SSv2-Split emphasizes everyday actions distinguished by spatiality, state changes and object interaction, while FineGym-Split targets a specialized domain with differences in body pose, motion and repetitions.

## 6 EXPERIMENTS & RESULTS

### 6.1 IMPLEMENTATION DETAILS.

Our base model uses ViT-Small (Dosovitskiy et al., 2021) pretrained with MVD (Wang et al., 2023) on Kinetics-400 (Carreira & Zisserman, 2017). We follow MVD's fine-tuning recipe for SSV2, training on 4 NVIDIA A100 GPUs with the same hyperparameters, except for a larger batch size

Table 1: **Statistics of Category Splitting Datasets.** We create meta-datasets SSV2-Split and FineGym-Split, each with two subsets (A and B). For each subset we report the number of categories and videos used to train the base model, the number of coarse categories eligible for splitting, the number of resulting fine-grained subcategories we can split into, the average number of videos used to evaluate generality and locality per category split as well as an example coarse-to-fine split.

| Benchmark | Subset | Base Model | | Category Splitting | | Evaluation | | Example |
| | | categories | videos | coarse | fine | generality | locality | |
| --- | --- | --- | --- | --- | --- | --- | --- | --- |
| SSv2-Split | A | 119 | 169K | 27 | 92 | 429 | 24347 | bending→{bending so it deforms, bending until it |
| | B | 121 | 169K | 27 | 94 | 403 | 24373 | breaks, trying to bend something unbendable} |
| FineGym-Split | A | 178 | 23K | 23 | 155 | 173 | 7988 | handspring forward→{handspring forward 1.5 turn, |
| | B | 162 | 23K | 19 | 143 | 204 | 7957 | handspring forward 1 turn, handspring forward no turn} |

of 18 for efficiency. We fine-tuning FineGym with the same setting. Low-shot fine-tuning uses AdamW (Loshchilov & Hutter, 2019) with learning rate $1 \times 10^{-3}$, weight decay $1 \times 10^{-3}$, and batch size 16. The learning rate follows cosine annealing, and training runs for up to 100 epochs with early stopping based on an exponential moving average (EMA) of training or validation loss ($\beta=0.95$, patience=5, $\delta=1 \times 10^{-3}$). For zero-shot methods, we adopt a CLIP ViT-L/14 text encoder (Radford et al., 2021). In modifier alignment, we use an MLP with one 384d hidden layer, trained using AdamW with a learning rate of $1 \times 10^{-3}$, cosine annealing, and batch size 10, for up to 100 epochs. Early stopping is applied with EMA of cosine similarity between predicted and reference embeddings. For more details refer to Appendix B.

## 6.2 EVALUATION METRICS.

We evaluate category splitting methods with two criteria: generality and locality. **Generality** measures the accuracy of the edited classifier $f'_\theta$ on distinguishing subcategories that coarse category $c$ has been split into:

$$\text{Generality} = \frac{1}{M} \sum_{j=1}^{M} \mathbb{1} \left[ f'_\theta(x_j) = y_j \right], \quad y_j \in \mathcal{S}^c, \tag{12}$$

**Locality** measures how well the edited classifier $f'_\theta$ preserves performance on non-target categories. It is defined as the ratio between the accuracy of $f'_\theta$ and that of the original classifier $f_\theta$ on the unchanged portion of the test set:

$$\text{Locality} = \frac{\sum_{i=1}^{N} \mathbb{1} \left[ f'_\theta(x_i) = y_i \right]}{\sum_{i=1}^{N} \mathbb{1} \left[ f_\theta(x_i) = y_i \right]}, \quad y_i \in \mathcal{Y} \setminus \{c\} \tag{13}$$

A locality of 1 indicates that the model edit has no negative impact on other categories. Each target category is split in isolation and results are averaged. Metrics are multiplied by 100 for readability.

## 6.3 COMPARATIVE RESULTS

Since category splitting is a new task, there are no existing methods. As a point of comparison, we build baselines using vision-language models (VLMs) as external modules, which provide a natural way to leverage semantic information for fine-grained recognition. For each video the base model classifies as the target coarse category, we predict a fine-grained subcategory by measuring similarity between the video embedding and the text embeddings of candidate fine-grained labels. We evaluate a range of VLMs, spanning general-purpose, fine-grained, and video-specific models: CLIP (Radford et al., 2021), FLAIR (Xiao et al., 2025), FG-CLIP (Xie et al., 2025), VideoCLIP-XL (Wang et al., 2024a), VideoPrism (Zhao et al., 2024), and InternVideo2 (Wang et al., 2024c).

The results on both SSv2-Split and FineGym-Split are summarized in Table 2. All VLMs achieve perfect locality by construction, since they are applied externally and do not modify model parameters. However, their generality remains low. On SSv2-Split there is little advantage to using video-text models, with VideoCLIP-XL, VideoPrism and InternVideo2 all performing worse than CLIP and FG-CLIP on subset B. On FineGym-Split video-text models are somewhat stronger with VideoPrism achieving 21.7% generality on subset A compared to CLIP's 12.1% and FG-CLIP's 19.4. In contrast, our method achieves substantially higher generality across both datasets and subsets. For example, it reaches 45.9% on SSv2-Split subset A and 34.2% on FineGym-Split subset A, while maintaining near-perfect locality. These results show that category splitting can be accomplished directly in video-only classifiers, and that exploiting the latent structure of the classifier is more effective than relying on VLM pretraining.

Table 2: **Comparative Zero-Shot Results**. Comparison with baseline methods on SSv2 and Fine-Gym. Our approach has much better generality than vision-language models

| Method | SSv2-Split | | | | | | FineGym-Split | | | | | |
| | Subset A | | | Subset B | | | Subset A | | | Subset B | | |
| | Gen. | Loc. | Mean | Gen. | Loc. | Mean | Gen. | Loc. | Mean | Gen. | Loc. | Mean |
|---|---|---|---|---|---|---|---|---|---|---|---|---|
| CLIP (Radford et al., 2021) | 27.6 | 100.0 | 63.8 | 30.7 | 100.0 | 65.4 | 12.1 | 100.0 | 56.1 | 7.2 | 100.0 | 53.6 |
| FLAIR (Xiao et al., 2025) | 30.6 | 100.0 | 65.3 | 28.4 | 100.0 | 64.2 | 18.0 | 100.0 | 59.0 | 11.6 | 100.0 | 55.8 |
| FG-CLIP (Xie et al., 2025) | 30.9 | 100.0 | 65.4 | 30.8 | 100.0 | 65.4 | 19.4 | 100.0 | 59.7 | 13.9 | 100.0 | 56.9 |
| VideoCLIP-XL (Wang et al., 2024a) | 28.6 | 100.0 | 64.3 | 29.9 | 100.0 | 64.9 | 18.0 | 100.0 | 59.0 | 8.2 | 100.0 | 54.1 |
| VideoPrism (Zhao et al., 2024) | 28.2 | 100.0 | 64.1 | 29.3 | 100.0 | 64.7 | 21.7 | 100.0 | 60.9 | 11.4 | 100.0 | 55.7 |
| InternVideo2 (Wang et al., 2024c) | 25.9 | 100.0 | 62.9 | 21.8 | 100.0 | 60.9 | 17.4 | 100.0 | 58.7 | 10.8 | 100.0 | 55.4 |
| Ours | 46.3 | 98.9 | 72.6 | 38.4 | 98.9 | 68.7 | 34.2 | 97.8 | 66.0 | 18.9 | 97.9 | 58.4 |

Table 3: **Zero-shot Ablation**. Our modifier retrieval and alignment greatly improve generality by mining modifier vectors from the video-only classifier.

| Method | Generality | Locality | Mean |
|---|---|---|---|
| Vision-Language Model | $27.6_{\pm 0.0}$ | $100.0_{\pm 0.0}$ | $63.8_{\pm 0.0}$ |
| Modifier Retrieval | $45.0_{\pm 0.0}$ | $98.9_{\pm 0.0}$ | $71.9_{\pm 0.0}$ |
| Modifier Alignment | $46.3_{\pm 0.9}$ | $98.9_{\pm 0.0}$ | $72.6_{\pm 0.5}$ |

Table 4: **One-Shot Finetuning Ablation**. Constraining updates to the extended head ($\theta'_{head}$) avoids catastrophic forgetting when training with little data, and initializing it with Knowledge Retrieval provides a stronger starting point that boosts generality without sacrificing locality.

| Components Finetuned | Initialization | Generality | Locality | Mean |
|---|---|---|---|---|
| **Full Data Finetuning** | | | | |
| $\theta_{head}, \theta'_{head}$ | coarse category | $86.7_{\pm 0.1}$ | $22.1_{\pm 18.8}$ | $54.4_{\pm 9.4}$ |
| $\theta'_{head}$ | coarse category | $86.7_{\pm 0.1}$ | $19.2_{\pm 0.1}$ | $52.9_{\pm 0.0}$ |
| **One-Shot Finetuning** | | | | |
| $\theta_{backbone}, \theta_{head}, \theta'_{head}$ | coarse category | $33.6_{\pm 1.9}$ | $0.0_{\pm 0.0}$ | $16.8_{\pm 1.0}$ |
| $\theta_{head}, \theta'_{head}$ | coarse category | $50.7_{\pm 1.8}$ | $96.4_{\pm 0.3}$ | $73.5_{\pm 0.9}$ |
| $\theta'_{head}$ | random | $45.0_{\pm 2.2}$ | $98.8_{\pm 0.1}$ | $71.9_{\pm 1.1}$ |
| $\theta'_{head}$ | coarse category | $48.4_{\pm 2.6}$ | $98.4_{\pm 0.1}$ | $73.4_{\pm 1.3}$ |
| $\theta'_{head}$ | modifier alignment | $52.8_{\pm 2.8}$ | $98.2_{\pm 0.1}$ | $75.5_{\pm 1.4}$ |

## 6.4 Ablation Study

We perform ablations on SSv2-Split subset A, averaging results over three runs.

**Zero-Shot Ablation.** Table 3 shows zero-shot category splitting under different variants of our approach. As a baseline, we directly apply the vision–language model that supplies the text-encoder for our method: videos the base model predicts as coarse category $c$ are reassigned to the fine-grained subcategory with highest video–text similarity. This achieves perfect locality since the base classifier remains unchanged. However, its low generality (27.6%) shows that pretrained vision-language embeddings are insufficient for capturing fine-grained distinctions. In contrast, our modifier retrieval substantially improves generality to 45.0% while maintaining high locality (98.9%), showing we can effectively re-purpose distinctions already encoded in the classifier without any new data. Adding modifier alignment yields a further gain in generality (+1.3%), enabling better fitting and improved generalization to new modifiers. Together, these results demonstrate that zero-shot category splitting is not only possible but surprisingly effective when exploiting the classifier's latent structure.

**One-Shot Finetuning Ablation.** Table 4 compares different fine-tuning strategies with limited training data. Updating the full model ($\theta_{backbone}, \theta_{head}, \theta'_{head}$) in a one-shot setting achieves moderate generality (33.6%) but destroys locality (0.0) as shared parameters are overwritten. Restricting updates to the head (($\theta_{head}, \theta'_{head}$) or ($\theta'_{head}$)) restores high locality (98.4) and boosts generality (48.4%), showing the need to isolate edits. Initializing the extended head with our zero-shot modifier alignment boosts generality (+4.4% over coarse category initialization, +7.8% over random), while maintaining locality, further indicating the effectiveness of our zero-shot approach. Notably, our one-shot approach outperforms full-data finetuning (75.5 vs 54.4 mean) as full-data finetuning creates a strong bias to the new classes, severely reducing locality.

**Base Model Pretraining.** Table 5 examines how category splitting varies across different model pretraining. Editing a model trained from scratch produces the weakest results, underscoring the importance of strong prior representations. Using the visual encoder of CLIP (Radford et al., 2021) improves performance, indicating that image–text pretraining provides useful compositional structure.

Table 5: **Base Model Pretraining**. Stronger pretraining helps, but our method is effective even on models trained from scratch.

| Base Model | Generality | Locality | Mean |
|---|---|---|---|
| From Scratch | $37.0_{\pm 1.0}$ | $97.7_{\pm 0.1}$ | $67.4_{\pm 0.5}$ |
| CLIP (Visual) | $38.2_{\pm 0.6}$ | $98.0_{\pm 0.1}$ | $68.1_{\pm 0.3}$ |
| VideoMAE | $42.9_{\pm 0.7}$ | $98.5_{\pm 0.0}$ | $70.7_{\pm 0.4}$ |
| MME | $42.6_{\pm 0.7}$ | $99.0_{\pm 0.0}$ | $70.8_{\pm 0.3}$ |
| SIGMA | $44.1_{\pm 1.3}$ | $99.0_{\pm 0.0}$ | $71.6_{\pm 0.6}$ |
| MVD | $46.3_{\pm 0.9}$ | $98.9_{\pm 0.0}$ | $72.6_{\pm 0.5}$ |

Table 6: **Text Encoders** from video-text models provide little benefit over CLIP.

| Text Encoder | Generality | Locality | Mean |
|---|---|---|---|
| **Text-only** | | | |
| RoBERTa | $40.9_{\pm 0.0}$ | $99.2_{\pm 0.0}$ | $70.0_{\pm 0.0}$ |
| **Image-Text** | | | |
| CLIP | $46.3_{\pm 0.9}$ | $98.9_{\pm 0.0}$ | $72.6_{\pm 0.5}$ |
| **Video-Text** | | | |
| InternVideo2 | $36.9_{\pm 1.8}$ | $99.5_{\pm 0.0}$ | $68.2_{\pm 0.9}$ |
| VideoCLIP-XL | $45.5_{\pm 1.3}$ | $98.9_{\pm 0.0}$ | $72.2_{\pm 0.6}$ |
| VideoPrism | $46.5_{\pm 0.2}$ | $98.8_{\pm 0.0}$ | $72.7_{\pm 0.1}$ |

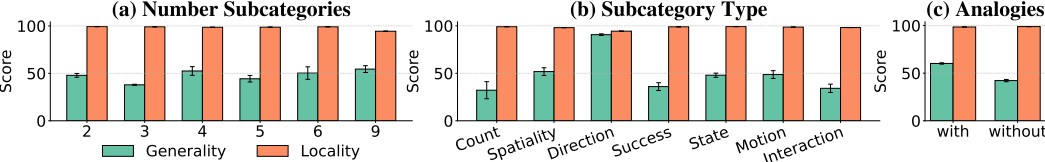

Figure 4: **Analysis over different category splits**. (a) Locality decrease slightly with more subcategories in the split, while generality shows no trend. (b) Performance is highest for direction-based splits and lowest for differences in object count, intent/success, and object interactions. (c) Existing analogous categories with the same modifier help, but our approach remains effective without them.

However, video-only pretraining proves most effective: VideoMAE (Tong et al., 2022), MME (Sun et al., 2023), SIGMA (Salehi et al., 2024) and MVD (Wang et al., 2023) all outperform CLIP. Among these models SIGMA and MVD achieve the strongest results, likely due to objectives that emphasize semantic structure over pixel-level reconstruction. Importantly, our method successfully edits all pretrained models, demonstrating robustness even when pretraining is limited.

**Text Encoder.** Table 6 compares different text encoders used in our zero-shot modifier alignment. Text-only model RoBERTa (Liu et al., 2019) achieves moderate generality (40.9%) but lags behind multimodal encoders. CLIP (Radford et al., 2021) improves generality (45.9%) and maintains locality, showing the benefit of visual alignment. However, video-text models do not consistently outperform CLIP: InternVideo2 (Wang et al., 2024c) performs poorly, while VideoCLIP-XL (Wang et al., 2024a) and VideoPrism (Zhao et al., 2024) achieve similar results. We thus conclude that multimodal alignment helps, but image-text models are sufficient for identifying relevant compositions.

## 6.5 ANALYSIS

We next examine where our model excels and where the room from improvement lies, with the aim of guiding future research in category splitting and fine-grained video understanding.

**Does performance degrade with more subcategories?** Figure 4a shows results across varying numbers of subcategories. We observe a slight decline in locality as the number of subcategories in the split increases, but generality has no clear trend. This suggests that while editing becomes somewhat less precise with more splits, the ability of the model to recognize new categories unaffected.

**Which subcategories are easier split into?** Figure 4b shows performance across subcategory types. Our model performs particularly well on direction as well as spatial position, motion and state change differences. Object count, action success and object interactions are the most difficult.

**Are subcategories with analogies easier?** Figure 4c compares subcategories where the relevant modifier is present in an existing category. As expected, performance is higher when analogies are available, confirming the compositional structure we discover in video classifiers. However, even without analogies, our method performs well, demonstrating its capacity for generalization.

**How much compositional variation is needed in the original label space?** To assess our method's dependence on the compositional variation in the original label space, Table 7 varies the proportion of coarse-grained categories used in the original label space before splitting. Our default setting used so far uses 50% of coarse categories. Increasing this to 66% and 75% reduces the number of fine-grained categories observed during training, thereby reducing the compositional structure available for modifier extraction. Despite this reduction generality decreases only slightly, while locality remains stable across all settings. Importantly, our method consistently outperforms CLIP regardless of how much compositional structure is present.

Table 7: **Effect of Compositional Variation in the Original Label Space.** Our method remains robust when reducing the number of fine-grained categories in the original label space and consistently outperforms CLIP.

| | Original Cat. | | | Cat. After Split | Ours | | | CLIP | | |
|---|---|---|---|---|---|---|---|---|---|---|
| % Coarse Cat. | Coarse | Fine | Total | Total | Gen. | Loc. | Mean | Gen. | Loc. | Mean |
| 50% | 27 | 92 | 119 | 174 | 46.3 | 98.9 | 72.6 | 27.6 | 100.0 | 63.8 |
| 66% | 36 | 67 | 103 | 174 | 44.2 | 99.1 | 71.7 | 29.0 | 100.0 | 64.5 |
| 75% | 41 | 55 | 96 | 174 | 44.6 | 98.8 | 71.7 | 29.4 | 100.0 | 64.7 |

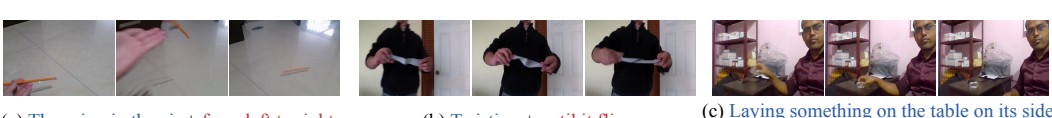

(a) Throwing in the air + from left to right    (b) Twisting + until it flips    (c) Laying something on the table on its side, not upright + in front of something

Figure 5: **Qualitative Results** on SSv2-Split with unseen category + modifier combinations.

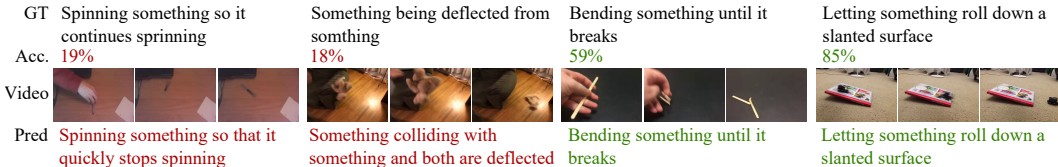

GT: Spinning something so it continues sprinning / Something being deflected from somthing / Bending something until it breaks / Letting something roll down a slanted surface

Acc.: 19% / 18% / 59% / 85%

Video

Pred: Spinning something so that it quickly stops spinning / Something colliding with something and both are deflected / Bending something until it breaks / Letting something roll down a slanted surface

Figure 6: **New Visual Distinctions**. We show failure and success cases of our model when categories require new visual distinctions.

### 6.6 QUALITATIVE RESULTS.

**Unseen Modifiers.** Figure 5 shows qualitative results on SSv2-Split. Our modifier alignment can successful classify videos with unseen category-modifier pairs (a) and completely unseen modifiers (b). Moreover, modifiers can be applied to to fine-grained categories to further subdivide them (c). **New Visual Distinctions**. Figure 6 shows examples that require genuinely new visual distinctions. These cases are more challenging, as our edits operate only on the classification head; the backbone must already encode the visual cues needed to support the new subcategories. For instance, concepts such as *continues* or *deflected* introduce visual distinctions absent from the original training setup, leading to failure cases. However, our method also succeeds on some edits that introduce previously unseen concepts, such as *breaks* or *slanted surface*. This suggests that certain fine-grained cues are implicitly captured by the backbone despite not being required by the original label space. These successes highlight the stronger compositional structure provided by more capable pretrained backbones such as MVD.

## 7 RELATED WORK

**Model Editing.** Model editing studies how to efficiently modify model behavior while preserving performance on untargeted inputs. Prior works follows three strategies: (i) adding external modules, such as retrievers or auxiliary networks (Mitchell et al., 2022b; Zheng et al., 2023); (ii) augmenting the model with extra parameters, like adapters or extra neurons (Yu et al., 2024; Huang et al., 2023); and (iii) intrinsic edits that directly update the model weights (Meng et al., 2022; Fang et al., 2025; Meng et al., 2024; Mitchell et al., 2022a). Most of this literature focuses on LLMs, but there is growing interest in vision, especially in image generation models (Bau et al., 2020; Arad et al., 2024; Orgad et al., 2023). Closest to us are works that edit image classifiers (Santurkar et al., 2021; Yang et al., 2024). Santurkar et al. (2021) modify prediction rules by mapping one existing model concept to another (e.g. 'snow'→'road') using a single exemplar and augmentations, while Yang et al. (2024) learn a hypernetwork to locate editable parameters without prior concept knowledge. Unlike prior work, which modifiers decision boundaries for existing labels to correct errors, we edit to expand the label space, replacing a coarse category with multiple fine-grained ones.

**Continual Learning.** Continual or class-incremental learning aims to learn new tasks and categories without forgetting prior knowledge. Approaches fall into three groups: (i) rehearsal, which trains the model with a mixture of old and new data to maintain past performance (Rolnick et al., 2019; Aljundi et al., 2019); (ii) regularization, which penalizes changes to parameters important for

old tasks (Kirkpatrick et al., 2017; Mitchell et al., 2018); and (iii) architectural, which expand or modify the network to assign capacity to new tasks (Mallya & Lazebnik, 2018; Aljundi et al., 2017). Few-shot variants address adding new classes from limited data (Tao et al., 2020; Zhao et al., 2023; Zhang et al., 2021a; Xiang et al., 2022). Unlike continual learning, which expands the label space with new categories, we split an existing category into finer subcategories. We address this in a zero-shot setting with no data from either existing or new categories.

**Fine-Grained Video Understanding.** Fine-grained video understanding aims to recognize subtle distinctions between visually and semantically similar actions (Goyal et al., 2017; Shao et al., 2020; Damen et al., 2018). Beyond action recognition, other works identify fine-grained differences in terms of skill (Doughty et al., 2018; Pirsiavash et al., 2014; Parmar & Tran Morris, 2017), repetitions (Hu et al., 2022; Zhang et al., 2020), adverbs (Doughty et al., 2020; Moltisanti et al., 2023), action attributes (Zhang et al., 2021b), action differences (Nagarajan & Torresani, 2024; Burgess et al., 2025), temporal prepositions (Bagad et al., 2023), chirality (Bagad & Zisserman, 2025; Price & Damen, 2019) or how and why actions happen (Perrett et al., 2025). Progress spans both video-only models (Tong et al., 2022; Sun et al., 2023; Wang et al., 2023) and video-text pretraining (Xu et al., 2021; Zhao et al., 2024; Wang et al., 2024c;a). Yet video-only classifiers assume fixed taxonomies, and video-text models require massive paired corpora but still struggle with fine-grained distinctions (Doughty et al., 2024; Thoker et al., 2025). We instead study category splitting in video-only classifiers, refining coarse labels into finer ones with minimal supervision while preserving original categories.

**Compositionality in Vision.** Compositionality has long been studied in computer vision, where complex categories built from simpler parts or attributes. Early work focused on part-based object recognition (Felzenszwalb et al., 2008; Fischler & Elschlager, 1973), evolving into compositional generation (Zhao et al., 2018; Tan et al., 2019) and interpretable representations (Böhle et al., 2022; Stone et al., 2017). Compositional reasoning has also been applied to human–object interaction detection (Hou et al., 2020; Kato et al., 2018) and compositional zero-shot learning (Misra et al., 2017; Nagarajan & Grauman, 2018), where models must recognize unseen combinations of known primitives. Large-scale vision–language models have renewed interest, supporting tasks like compositional retrieval (Baldrati et al., 2023; Hsieh et al., 2023) and visual editing (Ceylan et al., 2023; Kawar et al., 2023), with evidence that compositional structure emerges in their representations (Lewis et al., 2024; Trager et al., 2023; Berasi et al., 2025). Yet studies (Hsieh et al., 2023; Thrush et al., 2022; Tong et al., 2024; Yuksekgonul et al., 2023) show persistent weaknesses in object–attribute binding, spatial relationships, and other compositional inputs. While most prior work emphasizes compositionality in vision-language spaces, we focus on video-only classifiers and study how we can leverage compositional structure in such models to in category splitting.

## 8 DISCUSSION

**Summary.** We introduced the problem of category splitting: refining coarse categories into fine-grained subcategories with lightweight model edits. We showed that video classifiers already encode compositional knowledge that supports zero-shot editing of the classifier to create new categories without additional data. In the low-shot setting, fine-tuning only the extended head is effective with as little as one video per category, and performance improves further with our zero-shot initialization. On our new category splitting benchmarks, SSv2-Split and FineGym-Split, our method consistently outperforms VLM baselines, achieving higher generality while preserving locality.

**Future Work.** We show that video classifiers implicitly encode compositional structure: actions decompose into base concepts and modifiers that can be isolated and repurposed. Beyond category splitting, this compositionality enables interpretability, continual learning via incremental modifiers, and reasoning across contexts. Within category splitting, several avenues remain open. While restricting edits to the classification head preserves stability, it limits flexibility. Exploring edits deeper in the model could unlock richer adaptations, though maintaining locality will be challenging. Another step is increasing expressivity beyond single text-based modifiers to multiple modifiers, hierarchical taxonomies or alternative representations. The geometry of model also warrants study, as hyperbolic or spherical classifiers may better capture compositionality. Finally, extending category splitting to images, audio, or multimodal recognition will test its generality and broaden its impact.

**Acknowledgements.** This work is supported by the Dutch Research Council (NWO) under a Veni grant (VI.Veni.222.160).

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

# A    CATEGORY SPLITTING BENCHMARK DETAILS

## A.1    COARSE CATEGORY CONSTRUCTION

We provide the complete list of constructed coarse categories along with the fine-grained categories within each. Table 11 and Table 12 show our constructed category hierarchy for SSv2 and Fine-Gym288, respectively. Across both splits, it includes 174 fine-grained SSv2 categories, with 162 of them grouped into 54 coarse categories, while the rest remain ungrouped. For FineGym288, since the original data annotation has some repeated text label, for avoiding ambiguity, we remove 3 fine-grained categories which has the same text label with other fine-grained categories, and then operate grouping on the remaining 285 categories, with 271 of them grouped into 42 coarse categories, while the rest remain ungrouped.

## A.2    BENCHMARK SPLIT

For each benchmark split, we list the coarse categories merged in the original SSv2 dataset (Table 13, Table 14) and in FineGym (Table 15, Table 16).

# B    EXPERIMENT DETAILS

## B.1    BENCHMARK IMPLEMENTATION DETAILS

For benchmarking, we used the official implementations of all video-language models with default hyperparameters provided in the respective repositories. Specifically, for CLIP we used the ViT-L/14 model via the clip Python package; for VideoCLIP-XL we used the official implementation provided by the authors; for VideoPrism we used the videoprism_lvt_public_v1_large model; and for InternVideo2 we used the InternVideo2-clip-6B implementation. CLIP and VideoCLIP-XL experiments were run on a single A100 GPU, while VideoPrism and InternVideo2 experiments were run on a single H100 GPU.

## B.2    ADDITIONAL EXPERIMENTS

We present additional ablation studies on video backbone size, an analysis of text quality robustness for the zero-shot modifier alignment method, and an ablation on the modifier alignment method using different training pair compositions.

**Backbone Size.** Table 8 evaluates the effect of backbone capacity on zero-shot category splitting. Results are consistent across ViT-Small, ViT-Based and ViT-Large with mean performance varying only slightly (71.3-72.4). This suggest that even small models learn representations rich enough to support splitting categories into fine-grained subcategories, and additional capacity offers little benefit for this task.

Table 8: **Backbone Size Ablation**. Results are consistent across different backbone sizes.

| Method | Generality | Locality | Mean |
|---|---|---|---|
| ViT-Small | $46.3_{\pm 0.9}$ | $98.9_{\pm 0.0}$ | $72.6_{\pm 0.5}$ |
| ViT-Base | $43.6_{\pm 0.7}$ | $99.1_{\pm 0.0}$ | $71.3_{\pm 0.4}$ |
| ViT-Large | $45.0_{\pm 0.8}$ | $99.3_{\pm 0.0}$ | $72.1_{\pm 0.4}$ |

**Robustness of Text Encoder.** We assess the robustness of the text encoder used by our method in Table 9. We selected one coarse category *bending something* and introduced three types of perturbation to the modifier texts of its fine-grained categories: typos, multi-sentence descriptions, and multilingual labels (German and Chinese). Our method is reasonably robust to typos, while multi-sentence definitions introduce some performance drop, likely due to noise introduced from longer, more complex text. Multilingual labels cause the largest drop as CLIP's text encoder is optimized for English. Importantly, our framework is modular. The text encoder can be replaced with stronger or multilingual alternatives, allowing the approach to better support non-English labels and further broaden its applicability.

Table 9: **Robustness of Text Encoder**. Our method is reasonably robust to typos but struggles with multi-sentence and multi-lingual text.

| Text Perturbation | Generality | Locality | Mean |
|---|---|---|---|
| Clean | 51.3 | 99.4 | 75.4 |
| Typos | 48.9 | 99.4 | 74.2 |
| Multi-Sentence | 41.1 | 99.6 | 70.4 |
| Multi-Lingual (DE & ZH avg) | 27.3 | 99.2 | 63.2 |

**Ablation on Training Pair Composition.** Table 10 evaluates the effect of different training pair compositions on zero-shot modifier alignment. We compare training pair compositions with only modifier pairs ($\mathcal{D}_{mod}$) to those that also include existing-category and pseudo-coarse category pairs ($\mathcal{D}_{mod} \cup \mathcal{D}_{cat}$). The results consistently show that incorporating more diverse training pairs leads to improved performance, indicating that richer pair supervision helps the model better align modifiers and enhances zero-shot generalization.

Table 10: **Ablation on Training Pair Composition**. The results consistently show that enriching the training signal by incorporating more diverse pairs leads to improved performance.

| | SSv2-Split | | | | | | FineGym-Split | | | | | |
|---|---|---|---|---|---|---|---|---|---|---|---|---|
| | Subset A | | | Subset B | | | Subset A | | | Subset B | | |
| Training Pairs | Gen. | Loc. | Mean | Gen. | Loc. | Mean | Gen. | Loc. | Mean | Gen. | Loc. | Mean |
| $\mathcal{D}_{mod}$ | 44.9 | 99.3 | 72.1 | 40.2 | 99.2 | 69.7 | 27.6 | 98.5 | 63.1 | 14.0 | 98.5 | 56.2 |
| $\mathcal{D}_{mod} \cup \mathcal{D}_{cat}$ | 46.3 | 98.9 | 72.6 | 38.4 | 98.9 | 68.7 | 34.2 | 97.8 | 66.0 | 18.9 | 97.9 | 58.4 |

## B.3 ANALYSIS SETTING

We provide details on how we group different subcategory for analysis. Table 18 is the grouping for different subcategory types and Table 17 is the grouping for whether a subcategory has analogies.

## C STATEMENTS

### C.1 REPRODUCIBILITY STATEMENT

The source code and instructions for running all experiments will be publicly released upon paper acceptance. All experimental details necessary for reproducing the results are included in the main paper and appendix.

### C.2 THE USE OF LARGE LANGUAGE MODELS

We utilised a large language model (LLM) to assist with language polishing and minor grammatical corrections, as well as minor refinements to our grouping of sub-category types in Figure 4. All scientific content, experimental design, and conclusions were solely developed by the authors.

Table 11: Constructed category hierarchy for SSv2.

| Coarse categories | Fine-grained categories |
|---|---|
| Moving the camera and Turning the camera | Approaching something with your camera |
| | Moving away from something with your camera |
| | Turning the camera downwards while filming something |
| | Turning the camera left while filming something |
| | Turning the camera right while filming something |
| | Turning the camera upwards while filming something |
| Attaching something to something and Trying but failing to attach something to something because it doesn't stick | Attaching something to something |

*Continued on next page*

Table 11: Constructed category hierarchy for SSv2.

| Coarse categories | Fine-grained categories |
|---|---|
| | Trying but failing to attach something to something because it doesn't stick |
| Bending something | Bending something so that it deforms |
| | Bending something until it breaks |
| | Trying to bend something unbendable so nothing happens |
| Burying something in something and Digging something out of something | Burying something in something |
| | Digging something out of something |
| Closing something and Opening something | Closing something |
| | Opening something |
| Covering something with something and Uncovering something | Covering something with something |
| | Uncovering something |
| Dropping something with spatial relation | Dropping something behind something |
| | Dropping something in front of something |
| | Dropping something into something |
| | Dropping something next to something |
| | Dropping something onto something |
| Failing to put something into something because something does not fit | Failing to put something into something because something does not fit |
| Folding something and Unfolding something | Folding something |
| | Unfolding something |
| Hitting something with something | Hitting something with something |
| Holding something | Holding something |
| Holding something with spatial relation | Holding something behind something |
| | Holding something in front of something |
| | Holding something next to something |
| | Holding something over something |
| Putting something on the table | Laying something on the table on its side, not upright |
| | Putting something similar to other things that are already on the table |
| | Putting something that cannot actually stand upright upright on the table, so it falls on its side |
| | Putting something upright on the table |
| Letting something roll | Letting something roll along a flat surface |
| | Letting something roll down a slanted surface |
| | Letting something roll up a slanted surface, so it rolls back down |
| Lifting a surface with something on it | Lifting a surface with something on it but not enough for it to slide down |
| | Lifting a surface with something on it until it starts sliding down |
| Lifting something up completely | Lifting something up completely without letting it drop down |
| | Lifting something up completely, then letting it drop down |
| Lifting something with something on it | Lifting something with something on it |
| Lifting up one end of something | Lifting up one end of something without letting it drop down |
| | Lifting up one end of something, then letting it drop down |
| Moving part of something | Moving part of something |

*Continued on next page*

Table 11: Constructed category hierarchy for SSv2.

| Coarse categories | Fine-grained categories |
|---|---|
| Moving something across a surface | Moving something across a surface until it falls down |
| | Moving something across a surface without it falling down |
| Moving something and something | Moving something and something away from each other |
| | Moving something and something closer to each other |
| | Moving something and something so they collide with each other |
| | Moving something and something so they pass each other |
| Moving something | Moving something away from something |
| | Moving something away from the camera |
| | Moving something closer to something |
| | Moving something down |
| | Moving something towards the camera |
| | Moving something up |
| Picking something up and Pretending to pick something up | Picking something up |
| | Pretending to pick something up |
| Piling something up and Stacking number of something | Piling something up |
| | Stacking number of something |
| Plugging something into something and Plugging something into something but pulling it right out as you remove your hand | Plugging something into something |
| | Plugging something into something but pulling it right out as you remove your hand |
| Poking a hole | Poking a hole into some substance |
| | Poking a hole into something soft |
| Poking a stack of something | Poking a stack of something so the stack collapses |
| | Poking a stack of something without the stack collapsing |
| Poking something | Poking something so it slightly moves |
| | Poking something so lightly that it doesn't or almost doesn't move |
| | Poking something so that it falls over |
| | Poking something so that it spins around |
| Pouring something | Pouring something into something |
| | Pouring something into something until it overflows |
| | Pouring something onto something |
| | Pouring something out of something |
| | Pretending to pour something out of something, but something is empty |
| | Trying to pour something into something, but missing so it spills next to it |
| Pretending or failing to wipe something off of something and Wiping something off of something | Pretending or failing to wipe something off of something |
| | Wiping something off of something |
| Twisting something, Twisting (wringing) something wet until water comes out and Pretending or trying and failing to twist something | Pretending or trying and failing to twist something |

Table 11: Constructed category hierarchy for SSv2.

| Coarse categories | Fine-grained categories |
|---|---|
| | Twisting (wringing) something wet until water comes out |
| | Twisting something |
| Tearing something | Pretending to be tearing something that is not tearable |
| | Tearing something into two pieces |
| | Tearing something just a little bit |
| Pretending to close something without actually closing it and Pretending to open something without actually opening it | Pretending to close something without actually closing it |
| | Pretending to open something without actually opening it |
| Pretending to poke something | Pretending to poke something |
| Pretending to put something with spatial relation | Pretending to put something behind something |
| | Pretending to put something into something |
| | Pretending to put something next to something |
| | Pretending to put something on a surface |
| | Pretending to put something onto something |
| | Pretending to put something underneath something |
| Pretending to scoop something up with something and Scooping something up with something | Pretending to scoop something up with something |
| | Scooping something up with something |
| Pretending to spread air onto something and Spreading something onto something | Pretending to spread air onto something |
| | Spreading something onto something |
| Pretending to sprinkle air onto something and Sprinkling something onto something | Pretending to sprinkle air onto something |
| | Sprinkling something onto something |
| Pretending to squeeze something and Squeezing something | Pretending to squeeze something |
| | Squeezing something |
| Pretending to take something from somewhere and Taking something from somewhere | Pretending to take something from somewhere |
| | Taking something from somewhere |
| Pretending to take something out of something and Taking something out of something | Pretending to take something out of something |
| | Taking something out of something |
| Pretending to throw something and Throwing something | Pretending to throw something |
| | Throwing something |
| Pretending to turn something upside down and Turning something upside down | Pretending to turn something upside down |
| | Turning something upside down |
| Pulling something | Pulling something from behind of something |
| | Pulling something from left to right |
| | Pulling something from right to left |
| | Pulling something onto something |
| | Pulling something out of something |
| Pulling two ends of something | Pulling two ends of something but nothing happens |
| | Pulling two ends of something so that it gets stretched |
| | Pulling two ends of something so that it separates into two pieces |

Table 11: Constructed category hierarchy for SSv2.

| Coarse categories | Fine-grained categories |
| --- | --- |
| Pushing something | Pushing something from left to right |
| | Pushing something from right to left |
| | Pushing something off of something |
| | Pushing something onto something |
| | Pushing something so it spins |
| | Pushing something so that it almost falls off but doesn't |
| | Pushing something so that it falls off the table |
| | Pushing something so that it slightly moves |
| | Pushing something with something |
| Putting multiple things | Putting number of something onto something |
| | Putting something and something on the table |
| | Putting something, something and something on the table |
| Putting something with spatial relation | Putting something behind something |
| | Putting something in front of something |
| | Putting something into something |
| | Putting something next to something |
| | Putting something onto something |
| | Putting something underneath something |
| Putting something on a surface or onto something | Putting something on a flat surface without letting it roll |
| | Putting something on a surface |
| | Putting something on the edge of something so it is not supported and falls down |
| | Putting something onto a slanted surface but it doesn't glide down |
| | Putting something onto something else that cannot support it so it falls down |
| Putting something that can't roll onto a slanted surface | Putting something that can't roll onto a slanted surface, so it slides down |
| | Putting something that can't roll onto a slanted surface, so it stays where it is |
| Removing something, revealing something behind | Removing something, revealing something behind |
| Rolling something on a flat surface | Rolling something on a flat surface |
| Showing something's properties | Showing a photo of something to the camera |
| | Showing that something is empty |
| | Showing that something is inside something |
| Showing something with spatial relation | Showing something behind something |
| | Showing something next to something |
| | Showing something on top of something |
| Showing something to the camera | Showing something to the camera |
| Something colliding with something | Something being deflected from something |
| | Something colliding with something and both are being deflected |
| | Something colliding with something and both come to a halt |
| Something falling | Something falling like a feather or paper |
| | Something falling like a rock |
| Spilling something with spatial relation | Spilling something behind something |
| | Spilling something next to something |
| | Spilling something onto something |
| Spinning something | Spinning something so it continues spinning |
| | Spinning something that quickly stops spinning |

Table 11: Constructed category hierarchy for SSv2.

| Coarse categories | Fine-grained categories |
|---|---|
| Stuffing something into something | Stuffing something into something |
| Taking one of many similar things on the table | Taking one of many similar things on the table |
| Throwing something toward something | Throwing something against something |
| | Throwing something onto a surface |
| Throwing something in the air | Throwing something in the air and catching it |
| | Throwing something in the air and letting it fall |
| Tilting something with something on it | Tilting something with something on it slightly so it doesn't fall down |
| | Tilting something with something on it until it falls off |
| Tipping something over and Tipping something with something in it over, so something in it falls out | Tipping something over |
| | Tipping something with something in it over, so something in it falls out |
| Touching (without moving) part of something | Touching (without moving) part of something |

Table 12: Constructed category hierarchy for FineGym.

| Coarse categories | Fine-grained categories |
|---|---|
| (VT) round-off, flic-flac with 0.5 turn on, salto forward | (VT) round-off, flic-flac with 0.5 turn on, stretched salto forward with 1.5 turn off |
| | (VT) round-off, flic-flac with 0.5 turn on, stretched salto forward with 0.5 turn off |
| | (VT) round-off, flic-flac with 0.5 turn on, stretched salto forward with 1 turn off |
| | (VT) round-off, flic-flac with 0.5 turn on, stretched salto forward with 2 turn off |
| | (VT) round-off, flic-flac with 0.5 turn on, piked salto forward with 0.5 turn off |
| | (VT) round-off, flic-flac with 0.5 turn on, piked salto forward off |
| | (VT) round-off, flic-flac with 0.5 turn on, tucked salto forward with 0.5 turn off |
| (VT) round-off, flic-flac with 0.5 turn on, 0.5 turn to piked salto backward off | (VT) round-off, flic-flac with 0.5 turn on, 0.5 turn to piked salto backward off |
| (VT) round-off, flic-flac with 1 turn on, salto backward | (VT) round-off, flic-flac with 1 turn on, stretched salto backward with 1 turn off |
| | (VT) round-off, flic-flac with 1 turn on, piked salto backward off |
| (VT) round-off, flic-flac on, salto backward | (VT) round-off, flic-flac on, stretched salto backward with 2 turn off |
| | (VT) round-off, flic-flac on, stretched salto backward with 1 turn off |
| | (VT) round-off, flic-flac on, stretched salto backward with 1.5 turn off |
| | (VT) round-off, flic-flac on, stretched salto backward with 0.5 turn off |
| | (VT) round-off, flic-flac on, stretched salto backward with 2.5 turn off |
| | (VT) round-off, flic-flac on, stretched salto backward off |
| | (VT) round-off, flic-flac on, piked salto backward off |

*Continued on next page*

Table 12: Constructed category hierarchy for FineGym.

| Coarse categories | Fine-grained categories |
|---|---|
|  | (VT) round-off, flic-flac on, tucked salto backward off |
| (VT) tsukahara | (VT) tsukahara stretched with 2 turn |
|  | (VT) tsukahara stretched with 1 turn |
|  | (VT) tsukahara stretched with 1.5 turn |
|  | (VT) tsukahara stretched with 0.5 turn |
|  | (VT) tsukahara stretched salto |
|  | (VT) tsukahara stretched without salto |
|  | (VT) tsukahara piked |
|  | (VT) tsukahara tucked with 1 turn |
|  | (VT) tsukahara tucked |
| (VT) handspring forward on, salto forward | (VT) handspring forward on, stretched salto forward with 1.5 turn off |
|  | (VT) handspring forward on, stretched salto forward with 0.5 turn off |
|  | (VT) handspring forward on, stretched salto forward with 1 turn off |
|  | (VT) handspring forward on, piked salto forward with 0.5 turn off |
|  | (VT) handspring forward on, piked salto forward with 1 turn off |
|  | (VT) handspring forward on, piked salto forward off |
|  | (VT) handspring forward on, tucked salto forward with 0.5 turn off |
|  | (VT) handspring forward on, tucked salto forward with 1 turn off |
|  | (VT) handspring forward on, tucked double salto forward off |
|  | (VT) handspring forward on, tucked salto forward off |
| (VT) handspring forward on | (VT) handspring forward on, 1.5 turn off |
|  | (VT) handspring forward on, 1 turn off |
|  | (VT) handspring forward on, no turn off |
| (FX) leap with turn | (FX) switch leap with 0.5 turn |
|  | (FX) switch leap with 1 turn |
|  | (FX) split leap with 0.5 turn |
|  | (FX) split leap with 1 turn |
|  | (FX) split leap with 1.5 turn or more |
|  | (FX) johnson with additional 0.5 turn |
|  | (FX) switch leap to ring position with 1 turn |
|  | (FX) split leap with 1 turn or more to ring position |
|  | (FX) cat leap with 2 turn |
| (FX) leap | (FX) switch leap (leap forward with leg change to cross split) |
|  | (FX) split leap forward |
|  | (FX) johnson (switch leap with 0.25 turn to side split or to straddle pike position) |
|  | (FX) switch leap to ring position |
|  | (FX) split ring leap |
|  | (FX) stride leap forward with change of legs to wolf position |
|  | (FX) cat leap (leap with alternate leg change) |
| (FX) jump or hop with turn | (FX) split jump with 1 turn |
|  | (FX) split jump with 0.5 turn |

*Continued on next page*

Table 12: Constructed category hierarchy for FineGym.

| Coarse categories | Fine-grained categories |
| --- | --- |
| | (FX) split jump with 1.5 turn |
| | (FX) straddle pike or side split jump with 1 turn |
| | (FX) straddle pike or side split jump with 0.5 turn |
| | (FX) split jump with 1 turn or more to ring position |
| | (FX) stag jump with 0.5 turn |
| | (FX) tuck hop or jump with 1 turn |
| | (FX) tuck hop or jump with 2 turn |
| | (FX) stretched hop or jump with 1 turn |
| | (FX) pike jump with 1 turn |
| | (FX) wolf hop or jump with 1 turn |
| | (FX) hop with 0.5 turn free leg extended above horizontal throughout |
| | (FX) hop with 1 turn free leg extended above horizontal throughout |
| (FX) jump or hop | (FX) sissone jump (leg separation 180 degree on the diagonal to the floor, take off two feet, land on one foot) |
| | (FX) split jump (leg separation 180 degree parallel to the floor) |
| | (FX) straddle pike jump or side split jump |
| | (FX) stag ring jump |
| | (FX) ring jump (rear foot at head height, body arched and head dropped backward) |
| | (FX) stag jump |
| | (FX) sheep jump , jump with upper back arch and head release with feet almost touching head |
| | (FX) wolf hop or jump ( hop or jump with one leg bent and the other extended straight , forward above horizontal with knees together ) |
| | (FX) jump with legs separated, landing in front lying support |
| | (FX) butterfly forward, torso parallel to floor, slightly arched, legs straddled and feet above hip height during flight |
| (FX) 1 turn | (FX) illusion 1 turn through standing split |
| | (FX) 1 turn with free leg held upward in 180 split position throughout turn |
| | (FX) 1 turn in tuck stand on one leg, free leg optional |
| | (FX) 1 turn in back attitude, knee of free leg at horizontal throughout turn |
| | (FX) 1 turn on one leg, free leg optional below horizontal |
| | (FX) 1 turn with heel of free leg forward at horizontal throughout turn |
| (FX) 3 turn | (FX) 3 turn with free leg held upward in 180 split position throughout turn |
| | (FX) 3 turn in tuck stand on one leg, free leg straight throughout turn |
| | (FX) 3 turn on one leg, free leg optional below horizontal |
| (FX) 2 turn or more | (FX) 2 turn with free leg held upward in 180 split position throughout turn |
| | (FX) 2 turn in tuck stand on one leg, free leg straight throughout turn |

*Continued on next page*

Table 12: Constructed category hierarchy for FineGym.

| Coarse categories | Fine-grained categories |
|---|---|
| | (FX) 2 turn in back attitude, knee of free leg at horizontal throughout turn |
| | (FX) 2 turn on one leg, free leg optional below horizontal |
| | (FX) 2 turn or more with heel of free leg forward at horizontal throughout turn |
| (FX) 1 spin or less on back in kip position, hip and leg are closed | (FX) 1 spin or less on back in kip position, hip and leg are closed |
| (FX) 4 turn on one leg, free leg optional below horizontal | (FX) 4 turn on one leg, free leg optional below horizontal |
| (FX) take-off forward from one or both legs, salto sideward tucked and (FX) aerial cartwheel | (FX) take-off forward from one or both legs, salto sideward tucked |
| | (FX) aerial cartwheel |
| (FX) arabian double salto | (FX) arabian double salto tucked |
| | (FX) arabian double salto piked |
| (FX) salto forward | (FX) double salto forward tucked with 0.5 twist |
| | (FX) double salto forward tucked |
| | (FX) salto forward tucked |
| | (FX) double salto forward piked |
| | (FX) salto forward piked |
| | (FX) salto forward stretched with 2 twist |
| | (FX) salto forward stretched with 1 twist |
| | (FX) salto forward stretched with 1.5 twist |
| | (FX) salto forward stretched with 0.5 twist |
| | (FX) salto forward stretched, feet land successively |
| | (FX) salto forward stretched, feet land together |
| (FX) aerial walkover forward | (FX) aerial walkover forward |
| (FX) salto backward | (FX) double salto backward stretched with 2 twist |
| | (FX) double salto backward stretched with 1 twist |
| | (FX) double salto backward stretched with 0.5 twist |
| | (FX) double salto backward stretched |
| | (FX) salto backward stretched with 3 twist |
| | (FX) salto backward stretched with 2 twist |
| | (FX) salto backward stretched with 1 twist |
| | (FX) salto backward stretched |
| | (FX) salto backward stretched with 3.5 twist |
| | (FX) salto backward stretched with 2.5 twist |
| | (FX) salto backward stretched with 1.5 twist |
| | (FX) salto backward stretched with 0.5 twist |
| | (FX) double salto backward tucked with 2 twist |
| | (FX) double salto backward tucked with 1 twist |
| | (FX) double salto backward tucked |
| | (FX) salto backward tucked |
| | (FX) double salto backward piked with 1 twist |
| | (FX) double salto backward piked |
| | (FX) whip salto backward |
| (BB) ring and arch jump | (BB) yang-bo (jump to cross over split with body arched and head dropped backward) |
| | (BB) stag-ring jump |
| | (BB) ring jump (rear foot at head height, body arched and head dropped backward, 180 separation of legs) |
| | (BB) split ring jump (ring jump with front leg horizontal to the floor) |

*Continued on next page*

Table 12: Constructed category hierarchy for FineGym.

| Coarse categories | Fine-grained categories |
| --- | --- |
| | (BB) split ring leap |
| | (BB) sheep jump (jump with upper back arch and head release with feet to head height/closed Ring) |
| (BB) split and straddle jump | (BB) sissone (leg separation 180 degree on the diagonal to the floor, take off two feet, land on one foot) |
| | (BB) split jump with 0.5 turn in side position |
| | (BB) split jump with 0.5 turn |
| | (BB) split jump with 1 turn |
| | (BB) split jump |
| | (BB) straddle pike jump with 0.5 turn in side position |
| | (BB) straddle pike jump with 0.5 turn |
| | (BB) straddle pike jump with 1 turn |
| | (BB) straddle pike jump or side split jump in side position |
| | (BB) straddle pike jump or side split jump |
| | (BB) johnson with additional 0.5 turn |
| | (BB) johnson (leap forward with leg change and 0.25 turn to side split or straddle pike position) |
| (BB) split and switch leap | (BB) switch leap with 0.5 turn |
| | (BB) switch leap with 1 turn |
| | (BB) split leap with 1 turn |
| | (BB) switch leap (leap forward with leg change) |
| | (BB) stag split leap forward |
| | (BB) split leap forward |
| | (BB) switch leap to ring position |
| (BB) tuck hop or jump, pick jump and stretched hop or jump | (BB) tuck hop or jump with 1 turn |
| | (BB) tuck hop or jump with 0.5 turn |
| | (BB) pike jump from cross position |
| | (BB) stretched jump/hop with 1 turn |
| (BB) wolf hop or jump and cat leap | (BB) wolf hop or jump with 1 turn |
| | (BB) wolf hop or jump with 0.5 turn |
| | (BB) wolf hop or jump (hip angle at 45, knees together) |
| | (BB) cat leap (knees above horizontal alternately) |
| (BB) turn with legs in 180 split | (BB) 1 illusion turn through standing split, 180 legs separation |
| | (BB) 1.5 turn with free leg held upward in 180 split position throughout turn |
| | (BB) 1 turn with free leg held upward in 180 split position throughout turn |
| (BB) turns with free leg at horizontal and turns with free leg optional below horizontal | (BB) 1.5 turn with heel of free leg forward at horizontal throughout turn |
| | (BB) 2 turn with heel of free leg forward at horizontal throughout turn |
| | (BB) 1 turn with heel of free leg forward at horizontal throughout turn |
| | (BB) 2 turn on one leg, free leg optional below horizontal |
| | (BB) 1.5 turn on one leg, free leg optional below horizontal |
| | (BB) 1 turn on one leg, free leg optional below horizontal |

*Continued on next page*

Table 12: Constructed category hierarchy for FineGym.

| Coarse categories | Fine-grained categories |
|---|---|
| | (BB) 1 turn on one leg, thigh of free leg at horizontal, backward upward throughout turn |
| (BB) turn in tuck | (BB) 2.5 turn in tuck stand on one leg, free leg optional |
| | (BB) 1.5 turn in tuck stand on one leg, free leg optional |
| | (BB) 3 turn in tuck stand on one leg, free leg optional |
| | (BB) 2 turn in tuck stand on one leg, free leg optional |
| | (BB) 1 turn in tuck stand on one leg, free leg optional |
| (BB) salto backward | (BB) jump forward with 0.5 twist and salto backward tucked |
| | (BB) salto backward piked |
| | (BB) gainer salto backward stretched-step out (feet land successively) |
| | (BB) salto backward stretched-step out (feet land successively) |
| | (BB) salto backward stretched with legs together |
| (BB) salto sideward tucked | (BB) salto sideward tucked with 0.5 turn, take off from one leg to side stand |
| | (BB) salto sideward tucked, take off from one leg to side stand |
| (BB) free aerial | (BB) free aerial cartwheel landing in side position |
| | (BB) free aerial round-off |
| | (BB) free aerial cartwheel landing in cross position |
| | (BB) free aerial walkover forward, landing on one or both feet |
| (BB) arabian salto tucked (take-off backward with 0.5 twist, salto forward) | (BB) arabian salto tucked (take-off backward with 0.5 twist, salto forward) |
| (BB) salto forward tucked | (BB) salto forward tucked to cross stand |
| | (BB) salto forward tucked (take-off from one leg to stand on one or two feet) |
| (BB) salto forward piked to cross stand | (BB) salto forward piked to cross stand |
| (BB) flic-flac | (BB) flic-flac with 1 twist, swing down to cross straddle sit |
| | (BB) flic-flac, swing down to cross straddle sit |
| | (BB) gainer flic-flac, also with support on one arm |
| | (BB) flic-flac with 0.75 twist to side handstand, keep 2 seconds, lower to optional end position |
| | (BB) flic-flac with minimum 0.75 twist before hand support |
| | (BB) jump backward, flic-flac take-off with 0.5 twist through handstand to walkover forward, also with support on one arm |
| | (BB) flic-flac to land on both feet |
| | (BB) flic-flac with step-out, also with support on one arm |
| (BB) round-off | (BB) round-off |
| (BB) handspring forward with flight to land on one or both legs, also with support on one arm | (BB) handspring forward with flight to land on one or both legs, also with support on one arm |
| (BB) salto tucked | (BB) arabian double salto forward tucked |
| | (BB) salto forward tucked with 1 twist |

*Continued on next page*

Table 12: Constructed category hierarchy for FineGym.

| Coarse categories | Fine-grained categories |
| --- | --- |
| | (BB) salto forward tucked |
| | (BB) double salto backward tucked with 1 twist |
| | (BB) double salto backward tucked |
| | (BB) salto backward tucked with 1 twist |
| | (BB) salto backward tucked |
| | (BB) salto backward tucked with 1.5 twist |
| (BB) salto forward piked | (BB) salto forward piked |
| (BB) salto stretched | (BB) salto forward stretched with 1.5 twist |
| | (BB) salto forward stretched with 1 twist |
| | (BB) salto forward stretched |
| | (BB) salto backward stretched with 3 twist |
| | (BB) salto backward stretched with 2 twist |
| | (BB) salto backward stretched with 1 twist |
| | (BB) salto backward stretched |
| | (BB) salto backward stretched with 2.5 twist |
| | (BB) salto backward stretched with 1.5 twist |
| | (BB) salto backward stretched with 0.5 twist |
| (BB) double salto backward piked | (BB) double salto backward piked |
| (BB) gainer salto | (BB) gainer salto backward stretched with 1 twist to side of beam |
| | (BB) gainer salto tucked at end of beam |
| | (BB) gainer salto piked at end of beam |
| | (BB) gainer salto stretched with 1 twist at end of beam |
| | (BB) gainer salto stretched with legs together at end of the beam |
| (UB) circle backward with turn | (UB) pike sole circle backward with 1.5 turn to handstand |
| | (UB) pike sole circle backward with 1 turn to handstand |
| | (UB) pike sole circle backward with 0.5 turn to handstand |
| | (UB) giant circle backward with 1.5 turn to handstand |
| | (UB) giant circle backward with hop 1 turn to handstand |
| | (UB) giant circle backward with 1 turn to handstand |
| | (UB) giant circle backward with 0.5 turn to handstand |
| | (UB) clear hip circle backward with 1 turn to handstand |
| | (UB) clear hip circle backward with 0.5 turn to handstand |
| | (UB) clear pike circle backward with 1 turn to handstand |
| | (UB) clear pike circle backward with 0.5 turn to handstand |
| | (UB) stalder backward with 1 turn to handstand |
| | (UB) stalder backward with 0.5 turn to handstand |
| (UB) circle backward without turn | (UB) pike sole circle backward to handstand |
| | (UB) giant circle backward |
| | (UB) clear hip circle backward to handstand |
| | (UB) clear pike circle backward to handstand |
| | (UB) stalder backward to handstand |

Table 12: Constructed category hierarchy for FineGym.

| Coarse categories | Fine-grained categories |
|---|---|
| (UB) circle forward with turn | (UB) pike sole circle forward with 0.5 turn to handstand |
| | (UB) giant circle forward with 1 turn on one arm before handstand phase |
| | (UB) giant circle forward with 1 turn to handstand |
| | (UB) giant circle forward with 1.5 turn to handstand |
| | (UB) giant circle forward with 0.5 turn to handstand |
| | (UB) clear hip circle forward with 0.5 turn to handstand |
| | (UB) stalder forward with 0.5 turn to handstand |
| (UB) circle forward without turn | (UB) giant circle forward |
| | (UB) clear hip circle forward to handstand |
| | (UB) clear pike circle forward to handstand |
| | (UB) stalder forward to handstand |
| (UB) over high bar | (UB) (swing backward) with 0.5 turn and straddle flight backward over high bar to catch high bar |
| | (UB) counter straddle over high bar with 0.5 turn to hang |
| | (UB) counter straddle over high bar to hang |
| | (UB) counter piked over high bar to hang |
| | (UB) (pike sole circle backward) counter stretched (reverse hecht) in layout position over high bar to hang |
| (UB) hang on high bar | (UB) (swing backward or front support) salto forward straddled to hang on high bar |
| | (UB) (swing backward) salto forward piked to hang on high bar |
| | (UB) (swing forward or hip circle backward) salto backward with 0.5 turn piked to hang on high bar |
| | (UB) (swing backward) salto forward stretched to hang on high bar |
| | (UB) (swing forward) salto backward stretched with 0.5 turn to hang on high bar |
| (UB) transition flight | (UB) transition flight from high bar to low bar |
| | (UB) transition flight from low bar to high bar |
| (UB) (swing forward) double salto backward | (UB) (swing forward) double salto backward tucked with 1.5 turn |
| | (UB) (swing forward) double salto backward tucked with 2 turn |
| | (UB) (swing forward) double salto backward tucked with 1 turn |
| | (UB) (swing forward) double salto backward tucked |
| | (UB) (swing forward) double salto backward piked |
| | (UB) (swing forward) double salto backward stretched with 2 turn |
| | (UB) (swing forward) double salto backward stretched with 1 turn |
| | (UB) (swing forward) double salto backward stretched |
| (UB) (swing forward) salto with 0.5 turn into salto forward tucked | (UB) (swing forward) salto with 0.5 turn into salto forward tucked |

*Continued on next page*

Table 13: Merge Setting for SSv2 Split A.

| Merged Coarse Category |
| --- |
| Attaching something to something and Trying but failing to attach something to something because it doesn't stick |
| Bending something |
| Burying something in something and Digging something out of something |
| Closing something and Opening something |
| Dropping something with spatial relation |
| Letting something roll |
| Lifting a surface with something on it |
| Lifting something up completely |
| Moving something and something |
| Picking something up and Pretending to pick something up |
| Poking a stack of something |
| Twisting something, Twisting (wringing) something wet until water comes out and Pretending or trying and failing to twist something |
| Pretending to put something with spatial relation |
| Pretending to spread air onto something and Spreading something onto something |
| Pretending to squeeze something and Squeezing something |
| Pushing something |
| Putting multiple things |
| Putting something on the table |
| Putting something on a surface or onto something |
| Showing something with spatial relation |
| Something falling |
| Spinning something |
| Pretending to take something from somewhere and Taking something from somewhere |
| Tearing something |
| Throwing something in the air |
| Tilting something with something on it |
| Something colliding with something |

Table 12: Constructed category hierarchy for FineGym.

| Coarse categories | Fine-grained categories |
| --- | --- |
| (UB) salto forward tucked | (UB) (swing backward) double salto forward tucked |
| | (UB) (swing backward) double salto forward tucked with 0.5 turn |
| | (UB) (under-swing or clear under-swing) salto forward tucked with 0.5 turn |
| (UB) (swing backward) salto forward with 0.5 turn into salto backward tucked | (UB) (swing backward) salto forward with 0.5 turn into salto backward tucked |
| (UB) (swing forward) salto backward stretched with 2 turn | (UB) (swing forward) salto backward stretched with 2 turn |
| (UB) (swing forward) salto backward stretched | (UB) (swing forward) salto backward stretched |

Table 17: Grouping Detail of whether a subcategory has Analogies

| W or w/o Analogies | Subcategory |
| --- | --- |
| with | Trying to bend something unbendable so nothing happens |
| | Dropping something behind something |
| | Dropping something in front of something |
| | Dropping something into something |

Table 17: Grouping Detail of whether a subcategory has Analogies

| W or w/o Analogies | Subcategory |
|---|---|
| | Dropping something next to something |
| | Dropping something onto something |
| | Lifting something up completely without letting it drop down |
| | Lifting something up completely, then letting it drop down |
| | Pretending to put something behind something |
| | Pretending to put something into something |
| | Pretending to put something next to something |
| | Pretending to put something onto something |
| | Pretending to put something underneath something |
| | Pushing something from left to right |
| | Pushing something from right to left |
| | Pushing something onto something |
| | Pushing something so that it slightly moves |
| | Showing something behind something |
| | Showing something next to something |
| | Pretending to be tearing something that is not tearable |
| without | Attaching something to something |
| | Trying but failing to attach something to something because it doesn't stick |
| | Bending something so that it deforms |
| | Bending something until it breaks |
| | Burying something in something |
| | Digging something out of something |
| | Closing something |
| | Opening something |
| | Letting something roll along a flat surface |
| | Letting something roll down a slanted surface |
| | Letting something roll up a slanted surface, so it rolls back down |
| | Lifting a surface with something on it but not enough for it to slide down |
| | Lifting a surface with something on it until it starts sliding down |
| | Moving something and something away from each other |
| | Moving something and something closer to each other |
| | Moving something and something so they collide with each other |
| | Moving something and something so they pass each other |
| | Picking something up |
| | Pretending to pick something up |
| | Poking a stack of something so the stack collapses |
| | Poking a stack of something without the stack collapsing |
| | Pretending or trying and failing to twist something |
| | Twisting (wringing) something wet until water comes out |
| | Twisting something |
| | Pretending to put something on a surface |
| | Pretending to spread air onto something |
| | Spreading something onto something |
| | Pretending to squeeze something |
| | Squeezing something |
| | Pushing something off of something |
| | Pushing something so that it almost falls off but doesn't |
| | Pushing something so that it falls off the table |
| | Pushing something with something |
| | Putting number of something onto something |
| | Putting something and something on the table |
| | Putting something, something and something on the table |
| | Laying something on the table on its side, not upright |
| | Putting something similar to other things that are already on the table |

Table 17: Grouping Detail of whether a subcategory has Analogies

| W or w/o Analogies | Subcategory |
|---|---|
| | Putting something that cannot actually stand upright upright on the table, so it falls on its side |
| | Putting something upright on the table |
| | Putting something on a flat surface without letting it roll |
| | Putting something on a surface |
| | Putting something on the edge of something so it is not supported and falls down |
| | Putting something onto a slanted surface but it doesn't glide down |
| | Putting something onto something else that cannot support it so it falls down |
| | Showing something on top of something |
| | Something falling like a feather or paper |
| | Something falling like a rock |
| | Spinning something so it continues spinning |
| | Spinning something that quickly stops spinning |
| | Pretending to take something from somewhere |
| | Taking something from somewhere |
| | Tearing something into two pieces |
| | Tearing something just a little bit |
| | Throwing something in the air and catching it |
| | Throwing something in the air and letting it fall |
| | Tilting something with something on it slightly so it doesn't fall down |
| | Tilting something with something on it until it falls off |
| | Something being deflected from something |
| | Something colliding with something and both are being deflected |
| | Something colliding with something and both come to a halt |
| | Pushing something so it spins |

Table 18: Grouping Detail within Sub-Category Type

| Type | Subcategory |
|---|---|
| Count | Putting number of something onto something |
| | Putting something and something on the table |
| | Putting something, something and something on the table |
| Spatiality | Moving something and something away from each other |
| | Moving something and something closer to each other |
| | Dropping something next to something |
| | Showing something next to something |
| | Pretending to put something next to something |
| | Dropping something behind something |
| | Dropping something in front of something |
| | Showing something behind something |
| | Pretending to put something behind something |
| | Dropping something onto something |
| | Showing something on top of something |
| | Pretending to put something onto something |
| | Pretending to put something underneath something |
| | Pushing something onto something |
| | Pretending to put something on a surface |
| | Putting something on a surface |
| | Dropping something into something |
| | Pretending to put something into something |
| Direction | Pushing something from left to right |
| | Pushing something from right to left |

*Continued on next page*

Table 18: Grouping Detail within Sub-Category Type

| Type | Subcategory |
|------|-------------|
| Success | Trying but failing to attach something to something because it doesn't stick |
| | Attaching something to something |
| | Pretending to pick something up |
| | Picking something up |
| | Pretending or trying and failing to twist something |
| | Twisting something |
| | Pretending to spread air onto something |
| | Spreading something onto something |
| | Pretending to squeeze something |
| | Squeezing something |
| | Pretending to take something from somewhere |
| | Taking something from somewhere |
| | Pretending to be tearing something that is not tearable |
| | Trying to bend something unbendable so nothing happens |
| | Poking a stack of something without the stack collapsing |
| State | Bending something so that it deforms |
| | Bending something until it breaks |
| | Tearing something into two pieces |
| | Tearing something just a little bit |
| | Twisting (wringing) something wet until water comes out |
| | Burying something in something |
| | Digging something out of something |
| | Closing something |
| | Opening something |
| | Poking a stack of something so the stack collapses |
| Motion | Throwing something in the air and catching it |
| | Throwing something in the air and letting it fall |
| | Lifting a surface with something on it but not enough for it to slide down |
| | Lifting a surface with something on it until it starts sliding down |
| | Lifting up one end of something without letting it drop down |
| | Lifting up one end of something, then letting it drop down |
| | Pushing something so that it almost falls off but doesn't |
| | Pushing something so that it falls off the table |
| | Tilting something with something on it slightly so it doesn't fall down |
| | Tilting something with something on it until it falls off |
| | Letting something roll down a slanted surface |
| | Letting something roll up a slanted surface, so it rolls back down |
| | Letting something roll along a flat surface |
| | Putting something on a flat surface without letting it roll |
| | Pushing something so it spins |
| | Pushing something so that it slightly moves |
| | Spinning something so it continues spinning |
| | Spinning something that quickly stops spinning |
| Interaction | Moving something and something so they collide with each other |
| | Moving something and something so they pass each other |
| | Something being deflected from something |
| | Something colliding with something and both are being deflected |
| | Something colliding with something and both come to a halt |
| | Pushing something off of something |
| | Pushing something with something |
| | Laying something on the table on its side, not upright |
| | Putting something similar to other things that are already on the table |
| | Putting something that cannot actually stand upright upright on the table, so it falls on its side |
| | Putting something upright on the table |

Table 18: Grouping Detail within Sub-Category Type

| Type | Subcategory |
| --- | --- |
| | Putting something on the edge of something so it is not supported and falls down |
| | Putting something onto a slanted surface but it doesn't glide down |
| | Putting something onto something else that cannot support it so it falls down |
| | Something falling like a feather or paper |
| | Something falling like a rock |

Table 14: Merge Setting for SSv2 Split B.

| Merged Coarse Category |
| --- |
| Moving the camera and Turning the camera |
| Covering something with something and Uncovering something |
| Holding something with spatial relation |
| Folding something and Unfolding something |
| Lifting up one end of something |
| Moving something across a surface |
| Moving something |
| Piling something up and Stacking number of something |
| Plugging something into something and Plugging something into something but pulling it right out as you remove your hand |
| Poking a hole |
| Pouring something |
| Pretending or failing to wipe something off of something and Wiping something off of something |
| Pretending to close something without actually closing it and Pretending to open something without actually opening it |
| Pretending to scoop something up with something and Scooping something up with something |
| Pretending to sprinkle air onto something and Sprinkling something onto something |
| Pretending to turn something upside down and Turning something upside down |
| Pulling something |
| Pulling two ends of something |
| Putting something with spatial relation |
| Putting something that can't roll onto a slanted surface |
| Showing something's properties |
| Spilling something with spatial relation |
| Pretending to take something out of something and Taking something out of something |
| Pretending to throw something and Throwing something |
| Throwing something toward something |
| Tipping something over and Tipping something with something in it over, so something in it falls out |
| Poking something |

Table 15: Merge Setting for FineGym288 Split A.

| Merged Coarse Category |
| --- |
| (VT) round-off, flic-flac with 0.5 turn on, salto forward |
| (VT) round-off, flic-flac with 1 turn on, salto backward |
| (VT) handspring forward on, salto forward |
| (FX) leap |
| (FX) jump or hop |
| (FX) 2 turn or more |
| (FX) 3 turn |
| (FX) take-off forward from one or both legs, salto sideward tucked and (FX) aerial cartwheel |
| (FX) salto forward |
| (FX) arabian double salto |
| (BB) ring and arch jump |
| (BB) wolf hop or jump and cat leap |
| (BB) turns with free leg at horizontal and turns with free leg optional below horizontal |
| (BB) salto backward |
| (BB) free aerial |
| (BB) salto stretched |
| (UB) circle backward without turn |
| (UB) circle forward with turn |
| (UB) hang on high bar |
| (UB) transition flight |
| (BB) salto forward tucked |
| (BB) gainer salto |
| (UB) (swing forward) double salto backward |

Table 16: Merge Setting for FineGym288 Split B.

| Merged Coarse Category |
| --- |
| (VT) round-off, flic-flac on, salto backward |
| (VT) tsukahara |
| (VT) handspring forward on |
| (FX) leap with turn |
| (FX) jump or hop with turn |
| (FX) 1 turn |
| (FX) salto backward |
| (BB) split and straddle jump |
| (BB) split and switch leap |
| (BB) tuck hop or jump, pick jump and stretched hop or jump |
| (BB) turn with legs in 180 split |
| (BB) turn in tuck |
| (BB) salto sideward tucked |
| (BB) flic-flac |
| (BB) salto tucked |
| (UB) circle backward with turn |
| (UB) circle forward without turn |
| (UB) over high bar |
| (UB) salto forward tucked |

