# OpenReview forum: "Let's Split Up: Zero-Shot Classifier Edits for Fine-Grained Video Understanding"
_ICLR.cc/2026/Conference — ICLR 2026 Poster_

### Official Review · Reviewer_8rDB · 2025-10-28

**Soundness:** 2
**Presentation:** 1
**Contribution:** 2
**Rating:** 4
**Confidence:** 4

**Summary:**

This paper introduces a new task named "category splitting", which aims to refine a coarse category into more detailed subcategories through classifier edits. They propose a zero-shot editing method that leverages the latent compositional structure within video models to create these new distinctions without requiring additional data. Besides, the study shows that low-shot fine-tuning is highly effective, and its performance is further enhanced when initialized with the edited weights. Experimental results on new benchmarks demonstrate that this approach significantly outperforms baselines, improving performance on the newly split categories while preserving performance on the others.

**Strengths:**

* This task introduces a new setting called "category splitting" and creates new benchmarks by reorganizing existing ones (SSv2, FineGym).
* The proposed method is reasonable and performs well on new splits of category, outperforming existing prevalent Vision-Language Models.
* The analysis section provides a sound analysis of the method's effectiveness, offering readers a deeper understanding of the method.

**Weaknesses:**

* The task setting proposed in this paper is akin to zero-shot classification or continual learning, but it is less challenging since the distributions of the fine-grained subcategories and the original coarse category are relatively close. Besides, a major limitation of this task setting is the requirement that the coarse and fine-grained categories must share the same base name, limiting the task's significance and practicality.
* The method section is hard to follow. There exist undefined or unclaimed symbols, and some statements are confusing.
* As stated in line 139, why use the mean of the associated fine-grained weight vectors as the pseudo vector of the coarse category? What if using the real text vector of coarse categories?
* According to Table 2, despite achieving sound results on the new categories, it still causes a drop on the other categories, whereas VLMs (e.g. CLIP) can avoid the problem.

**Questions:**

* Why not choose the image encoder from a VLM as the base model, as it may possess better latent compositional structure during the pre-training.

---

> ### Author Response · Authors · 2025-11-21
>
> We thank the reviewer for their time, effort and constructive feedback. We are glad to hear that the reviewer appreciated the new task, reasonable method, model performance and analysis. We address the reviewer’s remaining concerns below.
>
> ***
>
> **Task Setting and Limitations**
>
> We appreciate the reviewer’s comparison to zero-shot classification and continual learning. While our setting shares similarities with both, it is fundamentally distinct and not less challenging.
>
> - **Compared with continual learning:** continual learning adds entirely new, distinct categories and requires *labelled training data* for the new categories. In contrast, we split an existing category into finer subcategories *without any training examples* of those subcategories. This makes the problem decidedly challenging, as the model must disentangle subtle visual cues within a highly overlapping region of feature space, rather than learning from explicit supervision.
> - **Compared with zero-shot learning:** Zero-shot methods (e.g., VLMs and prototype-based models) are purpose-built to accept new categories at inference time and naturally support open-vocabulary expansion. In contrast, we aim to split categories in a fixed classifier that does not support label-set expansion. Introducing new subcategories therefore requires editing the classifier’s weights to create additional decision boundaries within an existing category. This is a different problem from applying a model that already supports zero-shot behavior, and not a simpler one.
>
> Since zero-shot methods operate under comparable supervision to our approach, we compare against them directly in Table 2.  In the revised paper, we also include two recent fine-grained VLMs: FLAIR (Xiao et al., 2025) and FG-CLIP (Xie et al., 2025). As shown in the updated table below, our method significantly outperforms all baselines in generality and overall performance.
>
> | Method                            | SSv2-Split Subset A Gen. | Loc.  | Mean | SSv2-Split Subset B Gen. | Loc.  | Mean | FineGym-Split Subset A Gen. | Loc.  | Mean | FineGym-Split Subset B Gen. | Loc.  | Mean |
> | --------------------------------- | ------------------------ | ----- | ---- | ------------------------ | ----- | ---- | --------------------------- | ----- | ---- | --------------------------- | ----- | ---- |
> | CLIP (Radford et al., 2021)       | 27.6                     | 100.0 | 63.8 | 30.7                     | 100.0 | 65.4 | 12.1                        | 100.0 | 56.1 | 7.2                         | 100.0 | 53.6 |
> | VideoCLIP-XL (Wang et al., 2024a) | 28.6                     | 100.0 | 64.3 | 29.9                     | 100.0 | 64.9 | 18.0                        | 100.0 | 59.0 | 8.2                         | 100.0 | 54.1 |
> | VideoPrism (Zhao et al., 2024)    | 28.2                     | 100.0 | 64.1 | 29.3                     | 100.0 | 64.7 | 21.7                        | 100.0 | 60.9 | 11.4                        | 100.0 | 55.7 |
> | InternVideo2 (Wang et al., 2024c) | 25.9                     | 100.0 | 62.9 | 21.8                     | 100.0 | 60.9 | 17.4                        | 100.0 | 58.7 | 10.8                        | 100.0 | 55.4 |
> | FLAIR (Xiao et al., 2025)         | 30.6                     | 100.0 | 65.3 | 28.4                     | 100.0 | 64.2 | 18.0                        | 100.0 | 59.0 | 11.6                        | 100.0 | 55.8 |
> | FGCLIP (Xie et al., 2025)         | 30.9                     | 100.0 | 65.4 | 30.8                     | 100.0 | 65.4 | 19.4                        | 100.0 | 59.7 | 13.9                        | 100.0 | 56.9 |
> | Ours                              | 46.3                     | 98.9  | 72.6 | 38.4                     | 98.9  | 68.7 | 34.2                        | 97.8  | 66.0 | 18.9                        | 97.9  | 58.4 |
>
> (Xiao et al., 2025) Rui Xiao, Sanghwan Kim, Mariana-Iuliana Georgescu, Zeynep Akata, Stephan Alaniz. "Flair: Vlm with fine-grained language-informed image representations." *Proceedings of the Computer Vision and Pattern Recognition Conference (CVPR)*. 2025.
>
> (Xie et al. 2025) Chunyu Xie, Bin Wang, Fanjing Kong, Jincheng Li, Dawei Liang, Gengshen Zhang, Dawei Leng, Yuhui Yin. "FG-CLIP: Fine-Grained Visual and Textual Alignment." *Proceedings of the International Conference on Machine Learning (ICML)*. 2025.

---

> ### Author Response · Authors · 2025-11-21
>
> **Requirement of Shared Base Name**
>
> We would like to clarify that our method does not require that coarse and fine-grained categories share the same base name.
>
> As shown in Table 10, coarse categories can be split into fine-grained subcategories with different verbs or even opposite meanings. For example, *closing something* vs. *opening something*, or *burying something in something* vs. *digging something out of something*. Many fine-grained actions naturally share a verb with their coarse parent,  with distinctions often being conveyed through additional verbs, prepositions, or adverbs. This reflects the structure of fine-grained video taxonomies, not a limitation of our method.
>
> Our method only requires that we can (1) group similar fine-grained categories to extract modifiers and (2) specify which modifiers we want to introduce when splitting a category. In the paper, we use text similarity as a practical mechanism because textual labels are available in most datasets and offer a convenient way to express these relationships. However, this choice is not inherent: the same procedure could use attribute similarity, metadata, or any structured descriptor of fine-grained variation without changing the method.
> ***
>
> **Clarity of the Method Section**
>
> We have revised the method section, ensuring all symbols are defined and confusing statements have been clarified.
> ***
>
> **Why Use Fine-Grained Averages for Coarse Categories**
>
> Our method operates in the classifier label space of a model trained only on videos, without any text. In this space, each fine-grained category has a learned weight vector, whereas its coarse parent category has no classifier weight vector, since the model was never trained on that label. To compare a fine-grained class to its coarse parent and extract the modifier direction, we approximate the coarse category using the mean of the weight vectors of its fine-grained members. This means reflects how the video-only classifier internally represents the coarse concept.
>
> Text is used only as a tool for organising and retrieving modifiers: (1) to group fine-grained classes when building the modifier dictionary, and (2) to retrieve the appropriate modifier when the user specifies a new subcategory. In both cases, text acts purely as an indexing mechanism. It does not interact with the classifier weights, nor does it influence the extraction of modifier vectors. Any structured metadata (e.g., attributes) could play the same role.
>
> Crucially, the text vector of the coarse category cannot replace the pseudo coarse vector in classifier space because it lies in a different embedding space, is not aligned with the classifier’s decision boundaries, and does not reflect the model’s own representation of the coarse concept. For this reason, averaging the fine-grained classifier weights is necessary for extracting modifier directions, while text remains a convenient and replaceable way to group and reference modifiers.
>
> ***
>
> **Impact on Other Existing Categories**
>
> We acknowledge that our editing method can cause a small performance drop on the original categories, whereas VLMs such as CLIP do not show this effect. However, this decrease is very limited and is offset by the substantial improvement on the newly created fine-grained subcategories. For example, our method reaches 46.3% generality on the new subcategories compared to 27.6% for CLIP, while the impact on the original categories is only a 1.1% reduction.

---

> ### Author Response · Authors · 2025-11-21
>
> **Image Encoder of CLIP as Backbone**
>
> As suggested by the reviewer, we add an experiment using the *image encoder from CLIP* as the base model to be edited. We add these results to our backbone ablation in Table 5, which is extended as shown below.
>
> | Base Model   | Generality | Locality   | Mean       |
> | ------------ | ---------- | ---------- | ---------- |
> | From Scratch | 37.0 ± 1.0 | 97.7 ± 0.1 | 67.4 ± 0.5 |
> | VideoMAE     | 42.9 ± 0.7 | 98.5 ± 0.0 | 70.7 ± 0.4 |
> | MME          | 42.6 ± 0.7 | 99.0 ± 0.0 | 70.8 ± 0.3 |
> | SIGMA        | 44.1 ± 1.3 | 99.0 ± 0.0 | 71.6 ± 0.6 |
> | MVD          | 46.3 ± 0.9 | 98.9 ± 0.0 | 72.6 ± 0.5 |
> | CLIP Visual Encoder   | 38.2 ± 0.6 | 98.0 ± 0.1 | 68.1 ± 0.3 |
>
> Using the CLIP visual encoder yields 38.2% generality, which is higher than all baselines in Table 2 (e.g., 30.9% for FGCLIP) and also higher than a model trained from scratch. This suggests that CLIP’s image–text pretraining does provide useful compositional structure for our task.
>
> However, the CLIP encoder performs below video-pretrained backbones such as VideoMAE, SIGMA and MVD. This is expected: video models benefit from temporal and motion cues that image encoders lack, and these cues appear important for capturing the fine-grained distinctions needed in category splitting.
>
> Overall, these results show that our method can successfully edit any trained classifier, image-pretrained, video-pretrained, or trained from scratch, while benefiting most from backbones whose training aligns with the fine-grained visual cues required by the split.

---

### Official Review · Reviewer_duDd · 2025-10-30

**Soundness:** 2
**Presentation:** 2
**Contribution:** 3
**Rating:** 6
**Confidence:** 4

**Summary:**

The paper proposes a method for zero-shot fine-grained classification by editing classifier weight matrices without retraining backbones. They extract "modifier vectors" by averaging coarse category weights from existing fine-grained examples and subtracting to isolate semantic differences, then use these to generate new subcategory weights via w_subcategory = w_coarse + v_modifier. The method is evaluated on SSV2-Split and FineGym-Split benchmarks, expanding classifier matrices from ~100 to ~150 categories while demonstrating improved performance over standard CLIP and VideoCLIP baselines on fine-grained classification tasks.

**Strengths:**

-- Practical & Elegant Solution: Addresses real problem of fine-grained classification without expensive retraining - just intelligent matrix manipulation

== Compositional Approach: The weight arithmetic (w_subcategory = w_coarse + v_modifier) is intuitive and enables systematic fine-grained category generation

**Weaknesses:**

-- Scalability Questions: Method requires existing fine-grained examples to extract modifiers, and matrix growth (100→150 categories) may not scale to truly large taxonomies

-- Clarity and Organization Issues: Paper was slightly difficult to follow on the main contribution and method - would benefit from restructuring for better readability (more intuitive figures perhaps?)

-- Insufficient Baseline Comparisons: Only compares against basic vision-language models (CLIP, VideoCLIP) rather than established fine-grained classification methods (e.g., other compositional approaches)

**Questions:**

I wonder, isn't it worth testing your model against fine-grained approaches (e.g., compositional approaches or methods that improve CLIP with additional losses for fine-grained evaluations)? There are many such methods, and while I'm not sure which one would be the best fit, I still raise this question. The current evaluation seems limited to basic vision-language baselines.

---

> ### Author Response · Authors · 2025-11-21
>
> We thank the reviewer for their time, effort and constructive feedback. We are glad to hear the reviewer appreciated the practicality and simplicity of our compositional editing approach. Below, we address the reviewer’s remaining concerns.
>
> ***
>
> **Scalability**
>
> It is true that our method requires existing fine-grained examples to extract modifiers. Importantly, however, our method also applies to modifiers that have no direct analogies in the original label space (see Fig. 4), demonstrating that it can scale beyond the specific modifiers present in the base model.
>
> To further assess scalability, we conducted an additional experiment on SSv2-Split. Since no larger taxonomy is available, we vary the proportion of coarse and fine-grained categories before splitting. This allows us to control (1) how many fine-grained classes are available for modifier extraction and (2) how large the category expansion is after splitting.
>
> | % Coarse Categories | Original Classes | Original Coarse Classes | Original Fine-Grained Classes | Total After Split | Ours (Gen. / Loc. / MEAN) | CLIP (Gen. / Loc. / MEAN) |
> | --------------- | ---------------- | ----------------------- | ----------------------------- | ----------------- | ------------------------- | ------------------------- |
> | 50%   | 119              | 27                      | 92                            | 174               | 46.3 / 98.9 / 72.6        | 27.6 / 100.0 / 63.8       |
> | 66%    | 103              | 36                      | 67                            | 174               | 44.2 / 99.1 / 71.7        | 29.0 / 100.0 / 64.5       |
> | 75%   | 96               | 41                      | 55                            | 174               | 44.6 / 98.8 / 71.7        | 29.4 / 100.0 / 64.7       |
>
> Even with fewer fine-grained classes to extract modifiers from and greater category growth after splitting, our method consistently outperforms the CLIP baseline and maintains strong generality and locality. This indicates that our approach remains effective with limited fine-grained supervision and is scalable to substantial category expansion. We have added this experiment to the revised paper in Table 7.
>
> ***
>
> **Clarity and Organisation**
>
> We agree with the reviewer and have revised the method section to improve clarity.

---

> ### Author Response · Authors · 2025-11-21
>
> **Fine-Grained Baselines**
>
> As suggested by the reviewer, we have added additional fine-grained methods for comparison. Specifically, we now include two state-of-the-art approaches, FLAIR (Xiao et al., 2025) and Fine-Grained CLIP (FGCLIP) (Xie et al., 2025), in the extended Table 2, shown below.
>
> | Method                            | SSv2-Split Subset A Gen. | Loc.  | Mean | SSv2-Split Subset B Gen. | Loc.  | Mean | FineGym-Split Subset A Gen. | Loc.  | Mean | FineGym-Split Subset B Gen. | Loc.  | Mean |
> | --------------------------------- | ------------------------ | ----- | ---- | ------------------------ | ----- | ---- | --------------------------- | ----- | ---- | --------------------------- | ----- | ---- |
> | CLIP (Radford et al., 2021)       | 27.6                     | 100.0 | 63.8 | 30.7                     | 100.0 | 65.4 | 12.1                        | 100.0 | 56.1 | 7.2                         | 100.0 | 53.6 |
> | VideoCLIP-XL (Wang et al., 2024a) | 28.6                     | 100.0 | 64.3 | 29.9                     | 100.0 | 64.9 | 18.0                        | 100.0 | 59.0 | 8.2                         | 100.0 | 54.1 |
> | VideoPrism (Zhao et al., 2024)    | 28.2                     | 100.0 | 64.1 | 29.3                     | 100.0 | 64.7 | 21.7                        | 100.0 | 60.9 | 11.4                        | 100.0 | 55.7 |
> | InternVideo2 (Wang et al., 2024c) | 25.9                     | 100.0 | 62.9 | 21.8                     | 100.0 | 60.9 | 17.4                        | 100.0 | 58.7 | 10.8                        | 100.0 | 55.4 |
> | FLAIR (Xiao et al., 2025)         | 30.6                     | 100.0 | 65.3 | 28.4                     | 100.0 | 64.2 | 18.0                        | 100.0 | 59.0 | 11.6                        | 100.0 | 55.8 |
> | FGCLIP (Xie et al., 2025)         | 30.9                     | 100.0 | 65.4 | 30.8                     | 100.0 | 65.4 | 19.4                        | 100.0 | 59.7 | 13.9                        | 100.0 | 56.9 |
> | Ours                              | 46.3                     | 98.9  | 72.6 | 38.4                     | 98.9  | 68.7 | 34.2                        | 97.8  | 66.0 | 18.9                        | 97.9  | 58.4 |
>
> As the results indicate, these fine-grained methods indeed outperform other VLM baselines, including dedicated video-language models. Nevertheless, our method still achieves substantially higher generality across all datasets while maintaining comparable locality, demonstrating its strength in introducing new fine-grained distinctions without degradation to existing categories.
>
> (Xiao et al., 2025) Rui Xiao, Sanghwan Kim, Mariana-Iuliana Georgescu, Zeynep Akata, Stephan Alaniz. "Flair: Vlm with fine-grained language-informed image representations." *Proceedings of the Computer Vision and Pattern Recognition Conference (CVPR)*. 2025.
>
> (Xie et al. 2025) Chunyu Xie, Bin Wang, Fanjing Kong, Jincheng Li, Dawei Liang, Gengshen Zhang, Dawei Leng, Yuhui Yin. "FG-CLIP: Fine-Grained Visual and Textual Alignment." *Proceedings of the International Conference on Machine Learning (ICML)*. 2025.

---

### Official Review · Reviewer_XTPn · 2025-11-01

**Soundness:** 3
**Presentation:** 3
**Contribution:** 4
**Rating:** 6
**Confidence:** 4

**Summary:**

This work introduces the task of category splitting for video recognition: starting from an already trained video classifier with a fixed label set, you later realize a coarse label like “poking something” or “dropping something” actually needs to be broken into several fine-grained subactions, but you don’t want to retrain the whole model or disturb performance on the other, unrelated classes. The paper’s key insight is that existing video models already contain reusable, compositional structure in their classification head: across related actions, the difference between labels often looks like a consistent “modifier” (e.g. direction, manner, target), so they build a modifier dictionary from existing labels and then synthesize new subcategory weights by adding an appropriate modifier vector to the original coarse class weight, producing new labels in a fully zero-shot way; to go beyond seen modifiers, they train a small alignment module that maps modifier text to modifier vectors, and if a few labeled clips for the new sublabels are available, they fine-tune only those new head weights to improve accuracy while keeping the rest of the model fixed, thus preserving locality. Evaluated on split versions of video datasets (like SSv2-Split and FineGym-Split), the method improves recognition of the new, finer labels without degrading performance on untouched classes, showing that you can “edit” a video model’s label space after training by exploiting the structured differences already present in its classifier.

**Strengths:**

- Proposes and explores a novel problem of clear practical relevance
- The work explores different forms of class splitting, both where the modifiers are seen or unseen
- The proposed solution is tidy and lightweight: they edit only the classifier head, reuse structure already present in the model by building a modifier dictionary from existing labels
- The authors construct 2 datasets for this problem from SSv2 and FineGym, which pose challenging test cases

**Weaknesses:**

- Method depends significantly on the assumption that the original label space already contains enough compositional variation to learn good modifier vectors

- Because the edit happens at the classifier head, it also assumes the backbone already captures the visual distinctions the new sublabels require; if the new split introduces visual novelty rather than just semantic refinement, a head-only edit will struggle, and the paper doesn’t really explore that failure mode

- There is some amount of over-claiming going on in this paper, e.g. the title claims "YOUR VIDEO MODEL CAN BE EDITED", but this method is fairly limited to a very specific set of situations.

**Questions:**

- You show that adding a retrieved/aligned modifier vector to the coarse class weight works, but how often is the best modifier actually coming from the same base action vs. being borrowed from a semantically different action? I would like to see some statistical analysis of this.

- How robust is the text encoder you use to messy, real-world label names (typos, multi-sentence definitions, multilingual labels etc) ? This choice should ideally be ablated

---

> ### Author Response · Authors · 2025-11-21
>
> We thank the reviewer for their time, effort and constructive feedback. We are glad to hear that the reviewer recognizes the novelty and practical relevance of our problem, our effort in building challenging datasets and the tidiness and lightweight nature of our solution. Below, we address the reviewer's remaining concerns.
>
> ***
>
> **Dependence on Compositional Variation in the Original Label Space**
>
> We agree with the reviewer that our method assumes a degree of compositional variation in the original label space. This assumption is stated on L107, and we have clarified it further in the revision. Importantly, however, our method also performs well for modifiers that do not have direct analogies in the original label space (see Fig. 4).
>
> To further quantify our model's dependence on compositional variation, we conducted an ablation on SSv2-split, varying the number of fine-grained categories in the original label space. In the paper, we use 50% of the available coarse-grained categories, keeping the remaining labels fine-grained (27 coarse / 92 fine-grained). Here, we also experiment with using 66% and 75% of coarse-grained categories, meaning there is less compositional variation in the original label space. These configurations result in 36 coarse/67 fine-grained and 41 coarse/55 fine-grained categories, respectively.
>
> | % Coarse Categories | Ours (Gen. / Loc. / MEAN) | CLIP (Gen. / Loc. / MEAN) |
> | ----------------- | ------------------------- | ------------------------- |
> | 50%              | 46.3 / 98.9 / 72.6        | 27.6 / 100.0 / 63.8       |
> | 66%              | 44.2 / 99.1 / 71.7        | 29.0 / 100.0 / 64.5       |
> | 75%              | 44.6 / 98.8 / 71.7        | 29.4 / 100.0 / 64.7       |
>
> As shown, reducing compositional variation leads to a small decrease in generality, however locality is maintained, and our method consistently outperforms the CLIP baseline across all settings. We have added this experiment to the revised paper in Table 7.
>
> ***
>
> **Dependence on Backbone for Visual Distinctions**
>
> We agree with the reviewer's observation: because our edit operates only on the classifier head, it assumes the backbone already encodes the visual distinctions required by the new sublabels. When a split introduces genuinely novel visual cues, a head-only edit can struggle. This is why our approach benefits from stronger pretrained backbones: for example, splitting categories in the MVD-pretrained model achieves 46.3% generality compared to 37.0% for the from-scratch model (Table 5).
>
> As requested, we now explicitly explore this failure mode. In the revised paper, we provide a qualitative analysis of four categories whose fine-grained variants require visual distinctions absent from the original label space (Figure 6). These examples illustrate that new visual distinctions are indeed a common failure case for head-only editing. For example, the model struggles with new fine-grained categories *spinning something so it continues spinning* and *something being deflected from something* as both *continues* and *deflected* are new visual concepts.  However, this analysis also shows that in some cases the backbone implicitly encodes fine-grained visual cues despite not being trained for them, enabling successful splits. For example, our approach has good accuracy for *bending something until it breaks* and *letting something roll down a slanted surface* despite *breaks* and *slanted surface* being unseen visual concepts.
>
> Our goal in this work is to introduce category splitting as a new problem and propose a simple yet effective first step toward addressing it. We expect that more expressive approaches, including editing or adapting the backbone, will be necessary for cases where new visual distinctions must be introduced, and we hope our analysis helps clarify these boundaries.
>
> ***
>
> **Paper Title**
>
> We will revise the title to *Let's Split Up: Zero-Shot Classifier Edits for Fine-Grained Video Understanding* to make it more precise.

---

> ### Author Response · Authors · 2025-11-21
>
> **Where the Best Modifier Comes From**
>
> We would like to clarify that, in our method, the retrieved modifier never comes from the same base action as the one being split. By design, the retrieval operates over *other* fine-grained classes in the label space, ensuring that the modifier is always borrowed from a semantically different action and allowing our approach to be zero-shot.
>
> For example, when splitting *holding something*, the modifier for *something in front of something* is retrieved from the existing category *"dropping something in front of something"*, not from any *holding* variant, as described in Section 3.1. Conversely, for *something over something*, no suitable modifier exists in the label space. As discussed in Section 3.2, modifier alignment is used to synthesis an appropriate modifier vector in such cases.
>
> Since retrieval from the same base action is impossible by construction, a statistical analysis would trivially yield 0% across all coarse categories. We have clarified this point more explicitly in the revised method description.
>
> ***
>
> **Robustness of Text Encoder to Noisy Labels**
>
> To assess the robustness of the text encoder used in our method (CLIP’s text encoder), we selected one coarse category, *bending something* and introduced three types of perturbation to the modifier texts of its fine-grained categories: typos,  multi-sentence descriptions, and multilingual labels (German and Chinese). The results are shown below.
>
> | Text Perturbation           | Gen. | Loc. | Mean |
> | ---------------------------- | ---- | ---- | ---- |
> | Clean                        | 51.3 | 99.4 | 75.4 |
> | Typos                        | 48.9 | 99.4 | 74.2 |
> | Multi-sentence definitions   | 41.1 | 99.6 | 70.4 |
> | Multilingual (DE & ZH avg) | 27.3 | 99.2 | 63.2 |
>
> Our method is reasonably robust to typos, while multi-sentence definitions introduce some performance drop, likely due to noise introduced from longer, more complex text.  Multilingual labels cause the largest drop as CLIP’s text encoder is optimized for English.
>
> Importantly, our framework is modular. The text encoder can be replaced with stronger or multilingual alternatives, allowing the approach to better support non-English labels and further broaden its applicability.

---

### Official Review · Reviewer_Lq1x · 2025-11-01

**Soundness:** 2
**Presentation:** 2
**Contribution:** 3
**Rating:** 4
**Confidence:** 2

**Summary:**

This paper presents a method for zero-shot adaptation of video-language models to perform fine-grained classification. This is tackled by "splitting" existing categories to sub-categories. The method improves over baselines. The paradigm is also extended to a few-shot setting where similar gains are shown.

**Strengths:**

The method shows good improvement over baselines. The authors also provide many ablations such as choice of encoder, pretraining among others. The dataset generated has also been provided fully aiding in transparency. The concept of zero shot adaptation to finer granularity levels is interesting and warrants more attention.

**Weaknesses:**

The paper uses the term "video model" very generally to refer specifically to a kind of video-language model. This is misleading as "video model" can refer to other concepts such as video generation models.

The method is very hard to follow as the authors do not provide any preliminary information of the architecture that they are based upon. Eg. it is hard to follow which are the weight vectors that are being referred to as additive.

Subjective: The title of the paper does not convey the problem being tackled.

**Questions:**

None

---

> ### Author Response · Authors · 2025-11-21
>
> We thank the reviewer for their time, effort and constructive feedback. We are glad to hear the reviewer appreciates the interesting nature of the problem, our improvement over baselines, comprehensive ablation studies, and dataset transparency. We address the remaining concerns below.
>
> ***
>
> **Clarification on "Video Model" and Paper Title**
>
> We agree with the potential confusion regarding the term *video model*. We have thus changed the title to *Let's Split Up: Zero-Shot Classifier Edits for Fine-Grained Video Understanding* for further clarity and have replaced the term *video model* with *video classifier*.
>
> ***
>
> **Clarification of the Base Architecture and Additive Weight Vectors**
>
> We design our method to work with any trained video classifier. This classifier can have any base architecture, as our approach only operates on the classification head. As noted on L268, by default, our experiments edit a ViT-Small classifier pretrained with MVD (Wang et al. 2023 on Kinetics-400 (Carreira & Zisserman, 2017). However, Table 5 demonstrates that our approach applies equally well to other backbones, including from-scratch, image-pretrained, and video-pretrained models.
>
> Regarding the *weight vectors referred to as additive*, these are the weights of the model's classification head. Our method leverages the observation that video classifiers exhibit natural compositional structure even without explicit training for it. We disentangle the weight components that encode fine-grained distinctions and add these vectors to the classifier weights of other coarse categories, enabling the creation of new fine-grained subcategory classifiers through a simple and interpretable edit.

---

### Comment · Area_Chair_sj5c · 2025-11-25
**Encourage discussion**

Hi all,

The authors have submitted their responses. Please take a moment to review them and see if they address your concerns.

Your thoughtful input is essential for a successful reviewing process and is greatly appreciated.

Many thanks,

Area Chair

---

### Meta-Review · Area_Chair_N83c · 2025-12-07

**Summary:**

This paper introduces category splitting for video recognition and studies how to refine coarse labels into finer subcategories through targeted classifier edits. The reviewers generally find the task meaningful and timely, and the proposed approach is lightweight and effective under both zero-shot and low-shot settings. The rebuttal improves the paper’s clarity and empirical positioning, making the overall contribution more convincing.

The initial scores were **4664**. The authors responded to the major concerns with clear clarifications and a more carefully framed presentation. The revision strengthens the presentation and better situates the work relative to relevant baselines and assumptions.

The revised version meets the standard for acceptance at ICLR, and I recommend acceptance.

**Reviewer Concerns:**

The rebuttal addressed concerns about task significance and positioning, methodological clarity, and evaluation completeness. The revised version presents a more balanced scope and a clearer argument for why category splitting is distinct from nearby settings and why the proposed editing strategy is a reasonable first step.

**Reviewer Scores:**

I expect Reviewer 8rDB may increase their rating after reading the rebuttal. Therefore, the final scores might be **4666**.

---

### Decision · Program_Chairs · 2026-01-26

Accept (Poster)